# Lightning-intense deep convective transport of water vapour into the UTLS over the Third Pole region

Prashant Singh and Bodo Ahrens

Institute for Atmospheric and Environmental Sciences, Goethe University Frankfurt, Frankfurt am Main, Germany

**Correspondence:** Prashant Singh (p.singh@iau.uni-frankfurt.de)

**Abstract.**

The Himalayas are known to be prominent locations for lightning-intense deep convective systems. Deep convective systems can transport significant amounts of water vapour into the upper troposphere and lower stratosphere (UTLS). Lightning data from the TRMM-LIS observation over 10 years, along with water vapour data from ERA5 reanalysis and satellite observations (AIRS, MLS), point to a possible link between the lightning-intense deep convective systems and water vapour in the UTLS over the Third Pole region. We used the climate model ICON-CLM at km-scale to investigate the transport of water vapour by lightning-intense deep convective systems. A year-long simulation indicates an increase in water vapour concentration during lightning events in the upper troposphere ($\sim$ 200 hPa). This finding is also supported by ERA5, AIRS, and MLS. Noticeably, ERA5 overestimates water vapour increases, especially during the monsoon period. A Lagrangian analysis of air parcels for over 1,600 lightning events, using ERA5 and ICON-CLM simulation, reveals that ERA5 transports considerably more air parcels to the upper troposphere than ICON-CLM simulation over the Third Pole region. The air parcels in the coarser-meshed ($\sim$ 30 km) convection-parametrised ERA5 data rise slowly, cross the Himalayas, and reach the upper troposphere over the Tibetan Plateau. In contrast, the km-scale convection-permitting ICON-CLM simulation shows fast vertical and less horizontal transport for the same events. In general, simulated lightning-intense deep convective events moisten the upper troposphere, but only a few instances may lead to direct moistening of the lower stratosphere over the Third Pole. Once an air parcel reaches the upper troposphere, its fate depends on synoptic circulation.

## 1 Introduction

Water vapour concentration varies significantly between the troposphere and the stratosphere. The troposphere contains much more water vapour, while the stratosphere remains relatively dry. The tropopause is the transition boundary between the troposphere and the stratosphere, which plays an essential role in water vapour exchange in the UTLS region (Zhang et al., 2023). Several mechanisms help to transport water vapour, trace gases, and even aerosols into the upper troposphere or lower stratosphere like deep convection, turbulent mixing, gravity waves, tropopause folding, etc. The water vapour in the UTLS absorbs outgoing long-wave radiation and affects the Earth's energy budget (Jain and Kar, 2017; Uma et al., 2014; Ueyama et al., 2023; Avery et al., 2017; Fueglistaler et al., 2005). Therefore, understanding the water vapour transport mechanism and quantifying the magnitude of UTLS exchange is essential.

Lightning is a clear indicator of deep convection because it occurs only in tall clouds (more than 4 km deep) with strong up-/downdrafts of supercooled water and ice crystals inside the cloud (Price, 2000; Singh et al., 2020; Brisson et al., 2021; Price et al., 2023; Singh and Ahrens, 2023). The early study by Price (2000) using extremely low frequencies (ELF) and satellite-based upper-tropospheric water vapour (UTWV) measurements found a strong correlation ($r = 0.9$) between global lightning frequency and UTWV. Furthermore, Price (2000) reported that continental deep convection significantly influences UTWV variation across different periods, unlike over oceanic regions. Follow up studies with TRMM lightning and World Wide Lightning Location Network (WWLLN) observations also indicated a strong empirical link between lightning and upper tropospheric water vapour (Price and Asfur, 2006; Price et al., 2023).

The Third Pole region, especially the Himalayan ranges, is classified as one of the global hotspot regions of lightning activity (Cecil et al., 2014; Singh and Ahrens, 2023). We see varying frequencies and intensities of lightning and thunderclouds from the west of the Himalayan mountains to the east of the mountain ranges. In the west of the Himalayan mountains, intense but less frequent lightning events, whereas in the east more frequent but less intense events were reported in Singh and Ahrens (2023). Various studies have identified multiple mechanisms driving water vapour transport between the lower stratosphere and upper troposphere over the Third Pole region, specially over the Tibetan Plateau. These include tropopause folds and enhanced planetary boundary layer dynamics in winter and deep convective transport driven by thermal heating and large-scale atmospheric processes in summer (Fadnavis et al., 2013; Škerlak et al., 2014; Jain and Kar, 2017; Hanumanthu et al., 2020; Clemens et al., 2023; Zhang et al., 2023). Previous studies over the Tibetan Plateau have used different definitions of tropopause for UTLS transport studies, including dynamical tropopause (Zhang et al., 2023; Škerlak et al., 2014), cold-point tropopause (Hanumanthu et al., 2020), and climatological tropopause height (Jain and Kar, 2017; Fadnavis et al., 2013).

There are several methods to define the tropopause height, depending on the objective of the study. For example, the thermal (WMO) tropopause is useful for operational and climatological purposes, the dynamical tropopause is applied in studies of UTLS exchange associated with jet streams, the chemical tropopause is important for examining ozone exchange, and the cold-point tropopause is particularly relevant for water vapour exchange studies (Pan et al., 2018; Zhang et al., 2023; Mehta et al., 2011). Usually, in the tropics, the cold-point tropopause is higher than the thermal (WMO) tropopause, whereas in the extratropics they are often at similar altitudes (Highwood and Hoskins, 1998; Mehta et al., 2011). The cold point tropopause controls the exchange of water vapour from the troposphere to the stratosphere by inhibiting most of the water vapour from crossing this lowest temperature region, where it undergoes freezing and dehydration (Schoeberl and Dessler, 2011; Fueglistaler et al., 2005; Fueglistaler and Haynes, 2005; Highwood and Hoskins, 1998). The Lagrangian cold-point tropopause, defined as the coldest temperature experienced along an air parcel's trajectory, can be higher than the Eulerian cold-point tropopause, which is determined from a fixed vertical grid. However, in some cases, the difference between the two may be small (Fueglistaler et al., 2005; Fueglistaler and Haynes, 2005).

Deep convective overshooting sometimes penetrates the cold-point tropopause and moisten the lower stratosphere, but such events are relatively infrequent as reported in previous observational studies over the United States (Avery et al., 2017; Ueyama et al., 2020; Jensen et al., 2020; Homeyer et al., 2023). Water vapour transported to the upper troposphere by convection can subsequently be lifted above the cold-point tropopause by later convective events, thereby bypassing the freezing–dehydration

constraint (Jensen et al., 2020; Ueyama et al., 2023; Avery et al., 2017; Homeyer et al., 2023). In addition, water vapour that has survived the freeze-drying process and remains in the upper troposphere can slowly ascend into the lower stratosphere through a large-scale ascent balanced by radiative heating (Jensen et al., 2020). Previous studies using backward and forward trajectory analyses above cloud tops have shown that the contribution of convection to lower stratospheric water vapour is $\sim 0.3$ ppmv globally, with slightly higher values near the Third Pole (Ueyama et al., 2023). Overall, deep convection shows limited potential to directly penetrate the lower stratosphere by crossing the cold-point tropopause; however, a moist upper troposphere can subsequently lead to increased lower stratospheric moisture through secondary processes.

Previous studies have primarily focused on tropical convection and its effects on the UTLS region (30 °S–30 °N) using low-resolution reanalysis and gridded satellite data. To better understand deep convective transport over the complex terrain of the Third Pole region, this study quantifies the influence of lightning-associated deep convective events on water vapour exchange between the lower troposphere to the UTLS using km-scale simulation. This paper is divided into four sections. The first section introduces the study and provides the necessary background. The second section describes the datasets used and the methodology for their analysis. The third section presents the results and key findings from the datasets and the km-scale ICON-CLM simulation. Finally, the fourth section summarizes the findings and offers recommendations based on this research.

## 2 Data and Methodology

In this section, we have provided detailed information on the satellite products used for the analysis of lightning events, as well as the reanalysis data employed in our study. We also outline the setup for the model simulations in detail.

### 2.1 Lightning Data

The following sections discuss satellite-observed lightning datasets used for both climatological patterns and specific event-based assessments in this study.

#### 2.1.1 TRMM

We have utilized two gridded lightning climatology datasets from the Tropical Rainfall Measuring Mission's (TRMM) - Lightning Imaging Sensor (LIS) and the Optical Transient Detector (OTD) observations (last accessed on: 10/04/2025; https://ghrc.nsstc.nasa.gov/pub/lis/climatology/LIS-OTD/). For the long-term analysis, we have used the daily gridded climatology data (LRTS) version 2.3 available at 2.5° x 2.5° from 1995 to 2015. This data is smoothed by applying a boxcar moving average (Cecil et al., 2014, 2017). Also, we have used monthly gridded data (HRMC) version 2.3, which is available at 0.5° x 0.5° resolution for the same period as daily data. This data set was also smoothed using the same method as daily data (Cecil et al., 2017).

### 2.1.2 ISS-LIS

The International Space Station (ISS) - LIS final quality controlled science product (not gridded) version 1 data was used for the period of this study 10/2019 – 09/2020 (last accessed on: 10/04/2025; https://ghrc.nsstc.nasa.gov/pub/lis/iss/data/science/final). ISS-LIS is the successor of TRMM-LIS, from 2016 onward ISS-LIS records the global lightning observation in total, i.e., inter-cloud and cloud-to-ground lightning (Blakeslee et al., 2014; Blakeslee, 2023). LIS from TRMM and ISS satellites observed the flash rate and radiant energy of the lightning events during the day and night. The ISS orbiting time is around 90 minutes; therefore we have observations over the Third Pole about 15-16 times per day.

The available gridded TRMM-LIS lightning dataset was used to understand the past trends in lightning activity over the Third Pole region and its association with enhanced water vapour transport to the UTLS region. This analysis serves as a foundational step in our study and provides a comparison with previous global lightning–upper troposphere water vapour studies (Price, 2000; Price and Asfur, 2006; Price et al., 2023). Since the TRMM-LIS mission was discontinued in 2015, we used ISS-LIS as its successor to analyse lightning activity in the study period. However, gridded ISS-LIS data are currently not available; only flash-level (latitude–longitude) observations are provided. Therefore, ISS-LIS was used to identify lightning events for km-scale simulation evaluation and convective transport analysis.

While TRMM-LIS and ISS-LIS cover different time periods, they both use identical LIS instrument, ensuring consistency in lightning detection. TRMM-LIS helps to establish the regional climatological context, whereas ISS-LIS supports specific event analysis. Since they are used for different but complementary purposes in this study long-term trend assessment and event-level validation, the temporal difference between the datasets does not impact the robustness of our study.

### 2.2 Water vapour

Due to the high operational cost of in-situ water vapour profile measurements, such observations are scarce and not easily accessible. Therefore, we used two sets of satellite data to analyse the climatology and perform a basic validation of model data.

### 2.2.1 AIRS

The Atmospheric Infrared Sounder (AIRS) aboard NASA's Aqua satellite collects infrared energy emitted by Earth's surface and atmosphere globally twice daily. In this study, we used water vapour data from level 3 products, i.e., quality-controlled level 2 data organized into 1° x 1° grid cells at various pressure levels, available on daily and monthly scales (last accessed on: 10/04/2025; https://disc.gsfc.nasa.gov/). For decade-long analysis, we used monthly gridded water vapour mixing ratio data (AIRS3STM) version 7, available from 2002 to 2024 (AIRS Science Team and Texeira, 2013). In addition, for the simulation evaluation, we used daily water vapour mixing ratio gridded data (AIRS3STD) version 7, available from 2002 to 2024.

### 2.2.2 MLS

Aura's Microwave Limb Sounder (MLS) instrument measures stratospheric temperatures and upper tropospheric constituents, such as water vapour, ozone, and sulfur dioxide, by detecting microwave emissions using a 190 GHz radiometer (last accessed on: 10/04/2025; https://disc.gsfc.nasa.gov/). The gridded level 3 MLS data is available at 4° x 5° resolution and various pressure levels from 2004 to 2024 (Lambert, 2021). In this study, like for AIRS, we have used MLS monthly binned water vapour mixing ratio (ML3MBH2O) version 5 for decade-long analysis. In addition, MLS's daily binned water vapour mixing ratio (ML3DBH2O) to evaluate the model simulation.

### 2.3 Reanalysis

ECMWF's fifth-generation atmospheric reanalysis (ERA5) data (Hersbach et al., 2020) at various pressure levels were used to analyse lightning climatology and lightning-intense deep convective events (last accessed on 10/04/2025; https://cds.climate.copernicus.eu/; DOI: 10.24381/CDs.adbb2d4). ERA5 provides hourly and monthly data for various atmospheric, oceanic, and land-surface quantities from 1940 to the present. ERA5 atmospheric reanalysis data is available at 31 km horizontal grid resolution and 137 vertical pressure levels up to 0.01 hPa.

In addition, we used Version 2 of the tropopause parameters derived from ERA5 reanalysis, available at the same spatial resolution and hourly temporal scale as ERA5 (last accessed on 10/04/2025; https://datapub.fz-juelich.de/slcs/tropopause/; DOI: 10.26165/JUELICH-DATA/UBNGI2). This dataset covers the study period from October 2019 to September 2020 and includes gridded data for the cold-point, dynamical, and the World Meteorological Organization (WMO)-defined tropopause heights (Hoffmann and Spang, 2021).

### 2.4 ICON-CLM

We used the numerical Icosahedral Nonhydrostatic (ICON) modelling framework in climate limited-area mode (CLM) (Van Pham et al., 2021). One-year-long simulation starting from 1st September 2019 and ending on 1st October 2020, using ICON-CLM version 2.6.4, were performed over the third pole region (covering 25ºN – 40ºN and 70ºE – 115ºE) with 3.3 km horizontal grid-spacing and 60 vertical levels (Singh and Ahrens, 2023). Initial condition and hourly lateral boundary updates were derived from ERA5. More details of this simulation are reported in the CORDEX-CPTP hydrological year studies Collier et al. (2024); Singh and Ahrens (2023), and summarised in Table 1. Here, the first simulation month was considered the model's spin-up time and was not analysed.

**Table 1.** Summary of the ICON-CLM set-up

| Model Setup | Scheme | Reference |
|---|---|---|
| Time step | 25 s | Singh and Ahrens (2023) |
| Boundary updates | 1 hr (ERA5) | Hersbach et al. (2020) |
| Deep convection | – | – |
| Shallow convection | Tiedtke | Tiedtke (1989) |
| Radiation | ecRad | Hogan and Bozzo (2016) |
| Cloud microphysics | single-moment | Doms et al. (2021) |
| Land surface process | TERRA | Schulz et al. (2016) |

## 2.5 Lagrangian Tracking

Air parcels during lightning events were tracked offline by integrating the advection equation in the Lagrangian form at each time step, as suggested in Miltenberger et al. (2013) based on LAGRANTO Lagrangian tracking model (Sprenger and Wernli, 2015). The four-dimensional wind data was used from the ICON-CLM km-scale simulation and ERA5 at hourly resolution. This offline Lagrangian model is available as python code developed by Hernández Pardo (2024) and can be accessed at https://github.com/lhpardo90/trajectory_calculator/ (last accessed on: 10/04/2025). This tool has been used in a previous study based on HALO flight observations with very high temporal resolution (Curtius et al., 2024).

## 3 Results and Discussion

### 3.1 Mean Tropopause Height over the Third Pole

The tropopause is the transition layer that separates the upper troposphere (UT) from the lower stratosphere (LS). It is not a sharply defined boundary, but rather a gradual transition zone. As a result, defining the sharp tropopause height can be challenging. In this study, we analysed tropopause heights from ERA5 reanalysis over the Third Pole region for the period from October 2019 to September 2020 (Hoffmann and Spang, 2021). The cold-point tropopause, defined as the altitude where temperature reaches its minimum between the troposphere and stratosphere. Based on the cold-point tropopause, the maximum tropopause height reaches $\sim 100$ hPa during September–October over the Third Pole region (Figure 1a). The average diurnal maximum tropopause height also shows a similar value around 100 hPa (Figure 1b). The dynamical tropopause, identified using a potential vorticity threshold, indicates an average maximum tropopause height in the range of 120–150 hPa between 25° and 30°N, on both monthly and diurnal timescales. The WMO thermal lapse rate definition shows the first tropopause height varying between approximately 150 hPa and 100 hPa, depending on time and location.

Since outputs from our km-scale model simulations are stored at relatively few pressure levels, calculating tropopause height using any one criterion can be misleading. Therefore, based on the range of tropopause heights derived from ERA5 using the three approaches, namely cold-point temperature, dynamical, and WMO definition (Figure 1), we have adopted 100 hPa as representative of the lower stratosphere (LS) and 200 hPa as the upper troposphere (UT) in our analysis. These pressure levels are consistently used in the following discussion as UT (200 hPa) and LS (100 hPa).

## (a) Max. Cold Tropopause Month

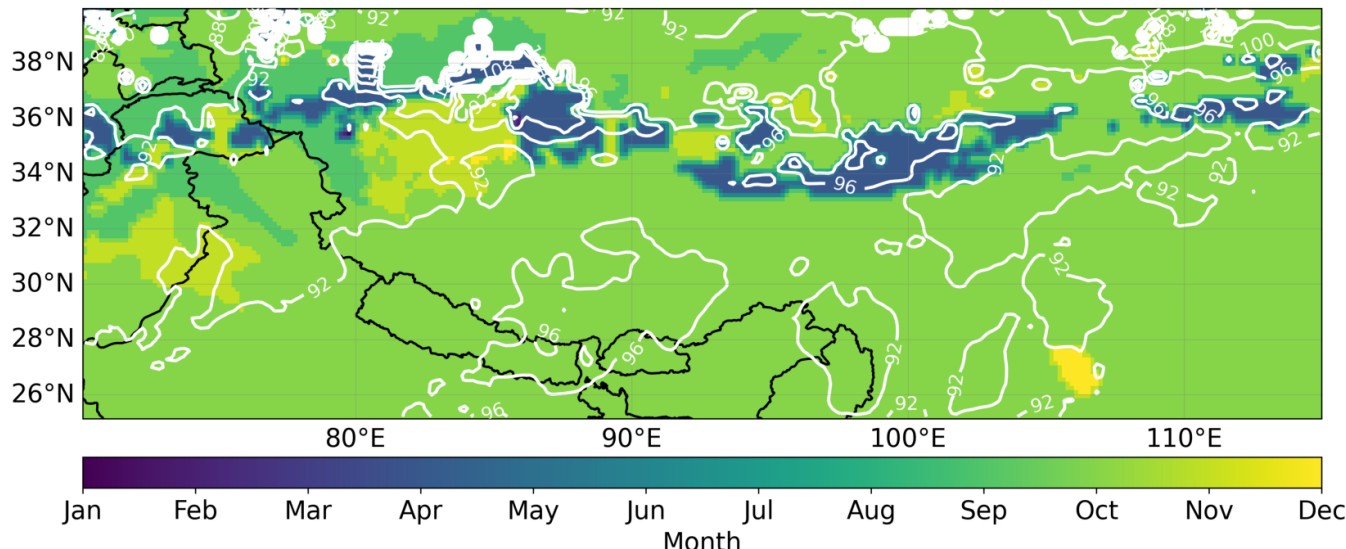

## (b) Max. Cold Tropopause Diurnal

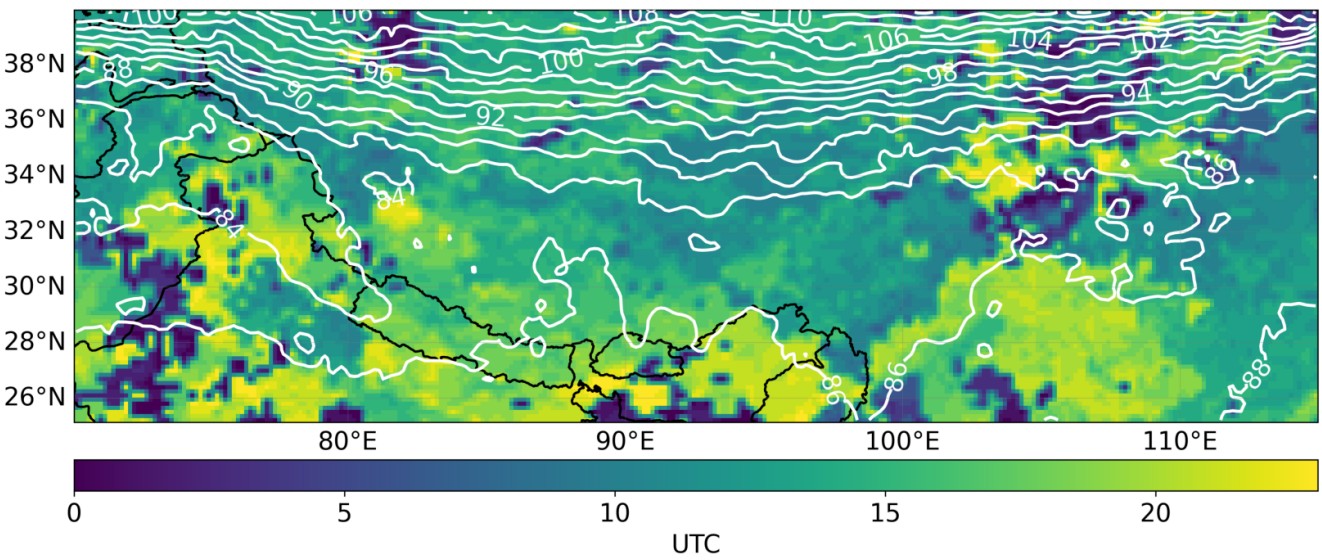

**Figure 1.** Maximum cold-point tropopause height derived from the local minimum temperature in ERA5 reanalysis. Contour lines indicate (a) the month of peak height and (b) the time of day when the height reaches its maximum, over the study region for October 2019–September 2020.

## 3.2 Climatological Overview

The TRMM lightning climatology identifies the Third Pole region as a global hotspot for lightning, with over 50 flashes/km$^2$/year reported in the western and eastern Himalayan regions (Cecil et al., 2014; Singh and Ahrens, 2023). Whereas the Tibetan Plateau shows the lowest density of lightning events (4 flashes/km$^2$/year) over the Third Pole (Figure A1). In this study, the Third Pole is defined to encompass the Tibetan Plateau, the Himalayan mountain range, and the Indo-Gangetic Plains. The subregions are explicitly outlined in figure A1. The analysis of TRMM gridded daily data from 1995 to 2014, along with specific humidity data from ERA5 for the same period, reveals a moderate correlation ($r = 0.66$) between daily lightning counts and upper tropospheric (UT) specific humidity at 200 hPa over the Third Pole region (Table 2). In contrast, the correlation coefficient between lightning counts and specific humidity at 100 hPa in the lower stratosphere (LS) region is lower. Despite the low frequency of observed lightning counts over the Tibetan Plateau, the correlation coefficient between lightning count and specific humidity in the upper troposphere and the lower stratosphere is highest in the study domain, followed by the Western Himalayan region (Table 2). Meanwhile, the Eastern Himalayan region with the highest annual lightning count shows no correlation with upper tropospheric moistening. Overall, this analysis suggests a moderate correlation between lightning events and water vapour in the UTLS over the Tibetan Plateau and Western Himalayan region. Although the average lightning density over the Tibetan Plateau is low (4 flashes/km$^2$/year), our correlation analysis focuses on a region (90–105°E, 30–35°N) where the accumulated lightning activity is substantial (Figure A1). During the TRMM-LIS observation period (1995–2014), this area recorded more than 3 million lightning flashes.

**Table 2.** Time series correlation ($r$) between daily lightning counts and ERA5 water vapour at 200 hPa and 100 hPa for different regions.

| Daily Lightning Counts | Water vapour | |
| --- | --- | --- |
| | **200 hPa** | **100 hPa** |
| Third Pole | 0.66 | 0.50 |
| Western Himalaya | 0.68 | 0.58 |
| Central Himalaya | 0.35 | 0.19 |
| Eastern Himalaya | 0.04 | -0.19 |
| Tibetan Plateau | 0.78 | 0.68 |

## 3.3 Decadal Overview

Since previous studies have reported biases in ERA5's upper tropospheric water vapour (Wright et al., 2025), we also conducted a long-term (2003 - 2013) monthly analysis of lightning events with satellite-derived water vapour over the Third Pole region. Figure 2a shows a scatter plot of lightning counts from a monthly TRMM data set with ERA5, AIRS, and MLS water vapour at the Upper Troposphere (200 hPa). Over the Third Pole domain, ERA5 specific humidity had a strong correlation coefficient ($r = 0.76$) with lightning counts, followed by MLS and AIRS. This temporal correlation between upper tropospheric moisture

and observed lightning events weakens as we investigate the lower stratosphere (100 hPa) over the Third Pole. Specifically, ERA5 and MLS's water vapour exhibit a weak correlation with lightning counts ($r < 0.42$), while AIRS shows a moderate correlation ($r = 0.56$) (Figure 2b). Since both lightning events and water vapour in the upper troposphere exhibit seasonal variability, the observed correlation may be influenced by seasonality. To address this, we applied a 12-month running average to the data presented in Figure 2. For ERA5 water vapour at 200 hPa, the correlation with lightning events was strong ($r = 0.77$), whereas at 100 hPa, the correlation was weaker. In contrast, the MLS and AIRS datasets showed weaker to moderate correlations at 100 hPa (Figure A2). The results after applying the 12-month running average are consistent with those shown in Figure 2 for ERA5 and AIRS. The 12-month running average of water vapour in the UTLS (Figure A2) suggests that ERA5 water vapour concentrations are not only higher than satellite observations in individual seasons (Figure 2), but are consistently higher across almost all seasons.

Over the Tibetan Plateau, water vapour in the upper troposphere shows a strong correlation ($r > 0.8$) with lightning events across all three datasets for the same period. However, in the lower stratosphere region, the correlation is moderate ($r > 0.6$), except for MLS, where it is 0.47 (Table A1). The Western Himalayan region also shows a moderate correlation in the upper troposphere as well as in the lower stratosphere. Central and Eastern Himalayan regions for the same period show weak or no correlation for the UTLS water vapour and lighting events. A previous study using radiosonde observations and MLS water vapour profiles have reported a wet bias in the upper troposphere and a dry bias in the stratosphere over the Tibetan Plateau (Yan et al., 2014). In contrast, AIRS data has been found to have a smaller bias in this region (Wang et al., 2017).

Monthly analysis of global lightning activity based on Schumann resonances and upper tropospheric water vapour (500–300 hPa) showed a strong correlation on the monthly timescale ($r = 0.85$), whereas the correlation decreased on the daily timescale ($r = 0.76$) (Price, 2000). Similarly, study by Price and Asfur (2006) over the African continent reported a significant daily correlation ($r = 0.76$) between lightning activity and upper tropospheric water vapour at 300 hPa. These findings suggest that lightning can serve as a proxy for upper tropospheric moistening over continental regions. For 2019, global gridded WWLLN lighting data set and specific humidity from ERA5 at 200 hPa shows ($r^2 = 0.72$) (Price et al., 2023). Which indicate significant relation between lightning and mid to upper tropospheric water vapour, which is quite evident upto upper troposphere (200 hPa) over the Third Pole region, and slightly low in the lower stratosphere (100 hPa) (Table 2, Figure 2). The differences between our findings and those of earlier studies can be attributed to two main factors. First, the spatial coverage differs: previous studies relied on globally averaged lightning activity inferred from Schumann resonances (Price, 2000), whereas our analysis uses TRMM-observed lightning data focused on a specific regional domain. The regional focus may reveal spatial heterogeneities that are smoothed out in global averages. Second, earlier studies examined water vapour over broader vertical layers such as 500–300 hPa (Price, 2000), up to 300 hPa (Price and Asfur, 2006), and 200 hPa (Price et al., 2023). In contrast, our study extends the analysis up to the upper troposphere and into the lower stratosphere (100 hPa), where specific humidity is naturally lower. Since specific humidity naturally decreases with altitude, the correlation between lightning activity and water vapour above 200 hPa becomes largely dependent on strong vertical transport processes. Supporting this, a study on tropical cyclones over the Indian subcontinent used lightning as an indicator of convection and reported that peak moistening during such intense storms occurs upto 200 hPa (Plotnik et al., 2021). Water vapour observations suggest that lightning may serve as a proxy for

upper tropospheric (200 hPa) moistening. However, in the lower stratosphere, this relationship appears weaker, particularly over the Third Pole region. Given the inherent biases and limitations of gridded observational datasets, high-resolution model simulations could provide deeper insights into the linkage between lightning activity and upper tropospheric moisture, as presented in the next subsection.

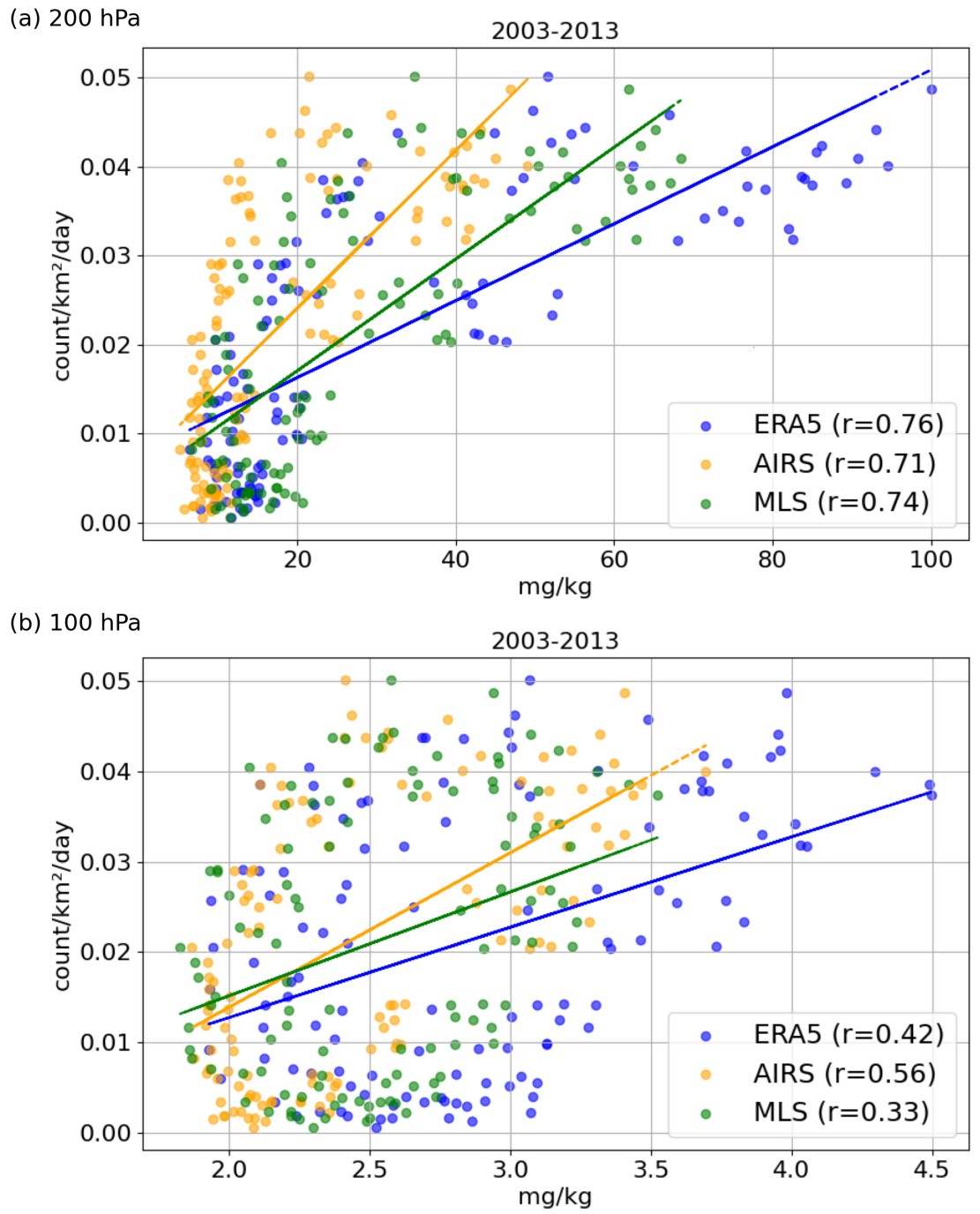

**Figure 2.** Scatter plots of monthly mean water vapour from satellite observations AIRS (yellow), MLS (green), and ERA5 reanalysis data (blue) at (a) 200 hPa and (b) 100 hPa against monthly lightning counts observed from TRMM during 2003-2013 over the Third Pole.

## 3.4 Annual Overview

A previous study by Singh and Ahrens (2023), using ICON-CLM simulations at km-scale (3.3 km) horizontal resolution with lateral ERA5 forcing, evaluated the model's performance in reproducing lightning events observed from ISS-LIS over the Third Pole region. That study reported that the ICON-CLM-simulated Lightning Potential Index (LPI) along with CAPE × Precipitation (CP) successfully predicted over 80% of the lightning events observed by ISS-LIS. The one-year-long ICON-CLM simulation and ISS-LIS lightning observations from October 2019 to September 2020 show a lightning hotspot over the Third Pole region similar to the long-term lightning climatology from TRMM (Cecil et al., 2014; Singh and Ahrens, 2023). During the study period, ISS-LIS observed more than 41000 lightning events over the Third Pole region; flash counts are more than 3 million. Since LPI shows good agreement with observed lighting events from ISS-LIS, in Figure 3, we have used LPI as the signature of lightning for the daily average over the domain. The daily ICON-CLM simulated LPI indicates high lightning activity over the Third Pole region during the pre-monsoon season (March–June), followed by the monsoon season, which is also well supported by TRMM-LIS and WWLLN observations. The pre-monsoon lightning peaks over the western and central Himalayan ranges are primarily driven by western disturbances, whereas those over the eastern Himalayan ranges are influenced by convection over the Bay of Bengal (Singh and Ahrens, 2023; Jadhav et al., 2025). Figure 3a shows daily water vapour at 200 hPa from ICON-CLM simulation, ERA5, AIRS, and MLS with LPI from ICON-CLM simulation. During winter and pre-monsoon seasons, all water vapour data sets showed comparable results. However, observations (AIRS, MLS) showed a lower water vapour concentration at 200 hPa during the monsoon season, while ERA5 showed a slightly higher concentration over the Third Pole region. Except for AIRS ($r = 0.66$), all other data sets show a strong correlation ($r = 0.73$) between LPI and upper tropospheric water vapour over the Third Pole region. Whereas in the lower stratosphere region (100 hPa), the correlation weakens for all the data sets ($r < 0.51$) (Figure 3b). This trend is consistent with the long-term correlation observed, as in Figures 2 from TRMM and water vapour gridded data, which showed a good correlation at 200 hPa and a weaker correlation at 100 hPa. It suggests that lightning-producing deep convective clouds can moisten the upper troposphere but are generally limited in reaching the lower stratosphere. Additionally, both ERA5 and ICON-CLM simulations consistently show higher water vapour concentrations in the UTLS compared to satellite observations (Figure 3). As previously discussed, the 12-month running average correlation reveals that ERA5 values are slightly shifted toward higher concentrations relative to satellite data, indicating a persistent positive bias in the UTLS throughout the year (Figure A2).

ICON-CLM simulated water vapour is significantly lower than ERA5 at 200 and 100 hPa during the monsoon season, yet still higher than satellite observations (Figure 3). One key reason is that ICON-CLM uses hourly boundary conditions from ERA5, resulting in a continuous high influx of moisture that drives the simulation toward a wet bias relative to satellite observations. During the monsoon period, mesoscale convective systems dominate, which allows large amounts of moisture to cross the atmospheric boundary layer (ABL) and enter the free troposphere (~500 hPa). As a result, not only ABL but the free troposphere also contains abundant moisture during monsoon season (Xu et al., 2022; Singh et al., 2020). However, due to the slow vertical transport of this moisture, it may introduce moist biases in the upper troposphere and lower stratosphere (UTLS) (Ploeger et al., 2024). As a result, the correlation between upper tropospheric water vapour and lightning activity is

lower during the monsoon than in the pre-monsoon season over the Third Pole region. Although LPI serves as an indicator

of lightning activity, it does not directly translate to observed lightning events. Therefore, to gain deeper insights, we have
analysed the seasonal effects and observed lightning occurrences in further detail.

(a) 200 hPa

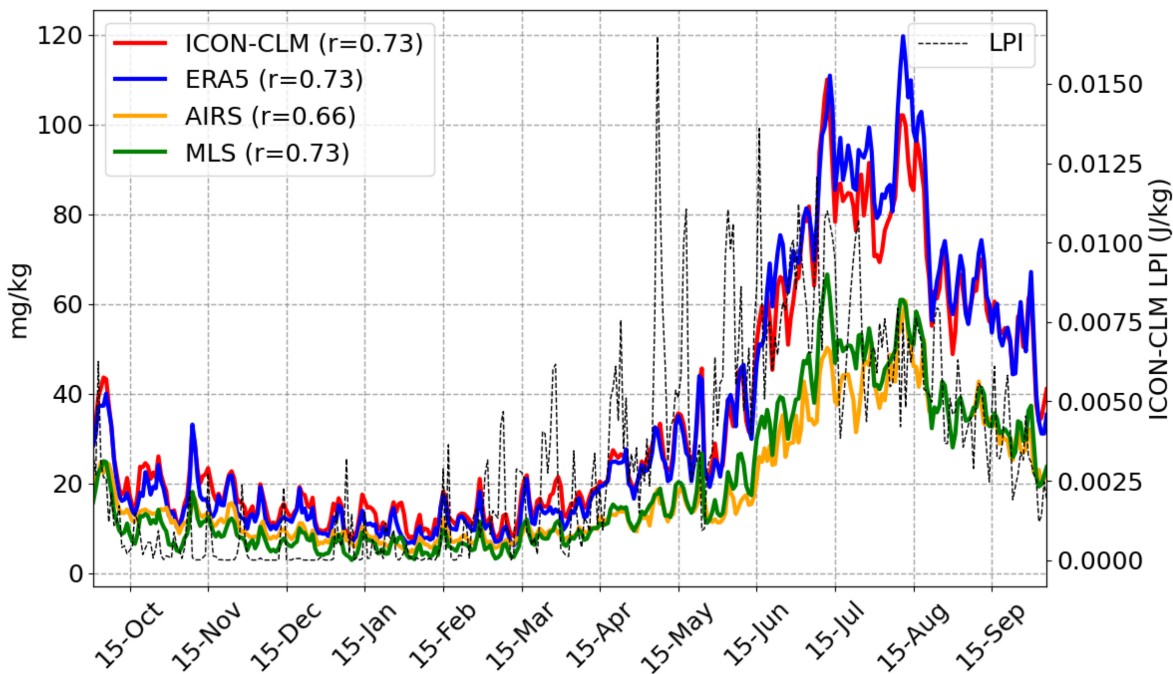

(b) 100 hPa

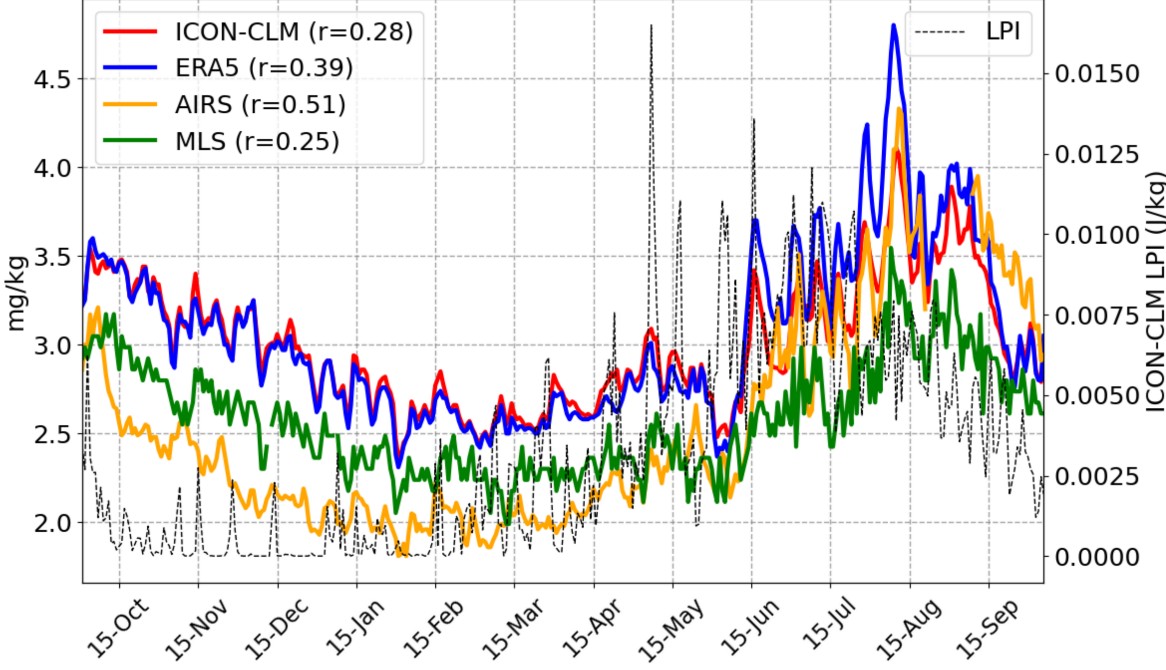

**Figure 3.** Time series of daily domain-averaged ICON-CLM simulated LPI (black) and water vapour from ICON-CLM simulation (red), ERA5 (blue), AIRS (yellow), MLS (green) at (a) 200 hPa and (b) 100 hPa.

Along with ICON-CLM simulated LPI and CP, we considered ISS-LIS and WWLLN observed lighting events for the study period over the Third Pole domain. Table 3 summarises the correlation ($r$) of water vapour in the upper troposphere and lower stratosphere from different data sets with respect to the strength of different lightning indicators (LPI, CP) and observations (ISS-LIS, WWLLN). During the pre-monsoon period, the LPI strongly correlates with water vapour in the upper troposphere (200 hPa), similar to the correlation observed on an annual scale. The LPI also exhibits a strong correlation with water vapour in the lower stratosphere (100 hPa) during pre-monsoon, in contrast to the annual pattern, where the correlation was as low as 0.3. During monsoon, LPI with water vapour from all the data sets shows a weak correlation in the upper troposphere and almost no correlation in the lower stratosphere. CP correlation with water vapour follows the same pattern: during monsoon the correlation is weak in the upper troposphere and negligible in the lower stratosphere, during pre-monsoon CP shows good correlation in the upper troposphere and lower stratosphere except for ERA5. Whereas the ISS-LIS observed events show a weak correlation in the upper troposphere and lower stratosphere during pre-monsoon, but also low or no correlation in the upper troposphere and lower stratosphere ($<0.2$) during monsoon. Since ISS-LIS observations are limited in time and space with ISS passes occurring every 90 minutes, they do not provide a complete picture of lightning activity throughout the entire day over the Third Pole region. Nevertheless, ISS-LIS also shows a higher correlation with water vapour in the UTLS region during the pre-monsoon season than in the monsoon season over the Third Pole region. Additionally, we used WWLLN daily gridded lightning density observations over the study period (Kaplan and Lau, 2021). The WWLLN results showed slight differences compared to LPI, CP, and ISS-LIS. Specifically, the correlation between lightning density and water vapour was weak in the upper troposphere and lower stratosphere ($r < 0.3$) during the pre-monsoon period. During the monsoon, the correlation ranged from weak to moderate across the upper troposphere and lower stratosphere ($r < 0.5$). As this dataset is not freely accessible for event-based lightning observations, we have not used it in our further analysis. Table 3 and Figure 3 show that during the pre-monsoon season, stronger water vapour transport occurs in the upper troposphere in association with lightning activity compared to the monsoon season. Additionally, CP and LPI, being gridded datasets with regular time steps, are consistent in both space and time. As a result, they offer more reliable comparisons compared to aggregated lightning observations, especially when compared with gridded water vapour data over the Third Pole region.

**Table 3.** Time series correlation between lightning indicators/observations and water vapour from various datasets (ICON-CLM, ERA5, AIRS, MLS) at 200 hPa and 100 hPa pressure levels during annual, pre-monsoon, and monsoon periods in the Third Pole region.

| Correlation Coefficient | 200 hPa | | | 100 hPa | | |
|---|---|---|---|---|---|---|
| | Annual | Pre-Monsoon | Monsoon | Annual | Pre-Monsoon | Monsoon |
| **LPI** | | | | | | |
| ICON-CLM | 0.73 | 0.73 | 0.37 | 0.28 | 0.62 | -0.03 |
| ERA5 | 0.73 | 0.72 | 0.36 | 0.39 | 0.55 | 0.08 |
| AIRS | 0.66 | 0.76 | 0.25 | 0.51 | 0.67 | 0.11 |
| MLS | 0.73 | 0.73 | 0.37 | 0.25 | 0.49 | 0.05 |
| **CP** | | | | | | |
| ICON-CLM | 0.83 | 0.72 | 0.43 | 0.37 | 0.59 | 0.03 |
| ERA5 | 0.83 | 0.70 | 0.42 | 0.49 | 0.21 | 0.10 |
| AIRS | 0.76 | 0.71 | 0.25 | 0.62 | 0.64 | 0.09 |
| MLS | 0.83 | 0.72 | 0.44 | 0.37 | 0.50 | 0.06 |
| **ISS-LIS** | | | | | | |
| ICON-CLM | 0.26 | 0.42 | -0.01 | 0.01 | 0.28 | -0.23 |
| ERA5 | 0.26 | 0.41 | -0.01 | 0.05 | 0.21 | -0.16 |
| AIRS | 0.24 | 0.40 | -0.08 | 0.17 | 0.37 | 0.03 |
| MLS | 0.26 | 0.39 | -0.01 | 0.03 | 0.18 | -0.25 |
| **WWLLN** | | | | | | |
| ICON-CLM | 0.45 | 0.30 | 0.33 | 0.23 | 0.13 | 0.54 |
| ERA5 | 0.45 | 0.25 | 0.35 | 0.29 | 0.09 | 0.51 |
| AIRS | 0.42 | 0.30 | 0.03 | 0.4 | 0.17 | -0.15 |
| MLS | 0.45 | 0.13 | 0.3 | 0.26 | 0.16 | 0.62 |

## 3.5 Annual Event Summary

Long-term climatology and daily observations indicate that lightning-associated transport processes can elevate water vapour into the upper troposphere over the Third Pole, particularly during the pre-monsoon season. Therefore, we examined individual events in different seasons and regions of the Third Pole to better understand which areas and processes are more likely to transfer water vapour to the UTLS region during lightning events. We selected 1,659 events detected by the ISS-LIS that corresponded to high-potential events of each month, based on the ICON-CLM simulation. The selection criteria were: (1) lightning flashes recorded by the ISS-LIS, and (2) simulated values of LPI ($>1$) and CP ($>10$), which were well-represented in the simulation. In addition, we selected 13 events over the south-central part of the Tibetan Plateau, as this region experiences the maximum lightning activity within the Tibetan Plateau (Figure A1). Including these additional events allows us to further investigate the role of high-elevation terrain in deep convective transport. Figure 4a shows the location of all the 1,672 lightning events selected that cover most of the Third pole and all the months of the study period. Most of the well-simulated lightning events are on the southern side of the Himalayan range, as per agreement with the climatology of the Third Pole's lightning activity.

For each lightning event (Figure 4a), 100 air parcels were initiated within a 1°grid box centred on the observed lightning event, spanning different pressure levels from the surface to 700 hPa (450 hPa for events over the Tibetan Plateau). The same set of initial points was used by both ERA5 and ICON-CLM meteorological data for the Lagrangian tracking of air parcels associated with each event. Lagrangian tracking with ICON-CLM simulated meteorology shows about 115 air parcels from 36 events reach the upper troposphere (above 300 hPa), including air parcels initiated at the Tibetan Plateau; the initial points of those air parcels are presented in Figure 4b. Most air parcels from the considered events reach the upper troposphere within the first few hours, particularly over the western and eastern Himalayan regions. While a few parcels ascend very quickly, the majority take less than 6 hours to reach the upper troposphere (Figure 4c). Very few air parcels may reach the lower stratosphere, as presented in Figure 4d. We have observed comparatively low-intensity lighting events over the Tibetan Plateau, and few of the air parcels reached the lower stratosphere.

The Lagrangian tracking using the aforementioned setup with ERA5 meteorology shows that more than a thousand air parcels from 54 events reached the upper troposphere, including a few from the Tibetan Plateau region (Figure 4e). During the observed lightning events, most air parcels took over 6 hours to reach the upper troposphere (Figure 4f). This indicates that ERA5 shows a slower ascent of air parcels to the upper troposphere than ICON-CLM for the observed deep convective (lightning) events. Except for the air parcels near the eastern part of the Himalayan Mountains, for ERA5, most of the air parcels show northward propagation in the upper troposphere during these events (Figure 4g), which is not that evident with ICON-CLM (Figure 4d).

We derive the Eulerian cold-point tropopause from ERA5 reanalysis and use it as a reference to track air parcels simulated with ERA5 and ICON-CLM simulated meteorology. Analysis shows that some parcels can reach near the cold-point tropopause but do not cross it (Figure A5). The Lagrangian cold-point tropopause height along individual air parcel trajectories can differ from the Eulerian value; however, this comparison is beyond the scope of the present study. In contrast, the thermal (WMO

lapse rate based) and dynamical tropopause heights derived from ERA5 indicate that many parcels can cross the tropopause and reach the lower stratosphere, whereas in ICON-CLM the number of such parcels is considerably smaller than ERA5. This clearly suggests that deep convective events can moisten the upper troposphere but rarely penetrate the cold-point tropopause to reach the lower stratosphere. The traditional thermal (WMO lapse rate based) and dynamical tropopause heights, typically used in aerosol–UTLS studies, indicate that air parcels driven by ERA5 meteorology can more frequently reach the lower

stratosphere, while those driven by ICON-CLM simulation do so for few events.

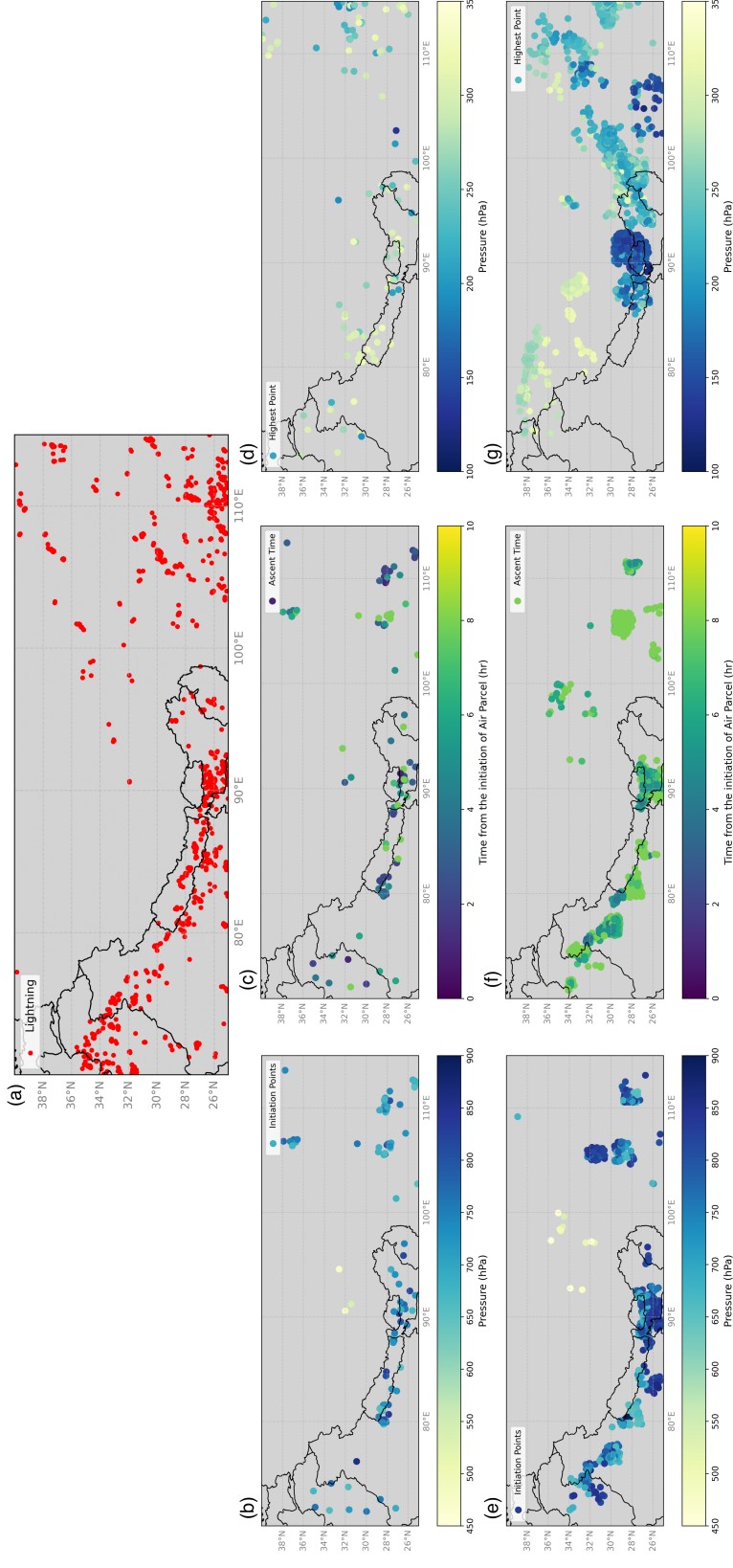

**Figure 4.** The figure compares air parcel tracking results for the lightning events using ICON-CLM and ERA5 meteorology. Panel (a) shows observed lightning event locations. Panels (b–d) display, for ICON-CLM, the initial air parcel locations, which are used as initial points in the lower troposphere for the forward Lagrangian tracking of air parcels. Panels (b–d) display, for ICON-CLM, the initial air parcel locations, time to reach upper troposphere (300 hPa), and final upper-tropospheric locations. Panels (e–g) present the same for ERA5.

Lightning-induced deep convective events simulated with ERA5 meteorology show a higher frequency of water vapour transport to the UTLS region than ICON-CLM simulation for the evaluated events (Figure 4, Figure A5). A monthly analysis of these events during the winter months (December, January, and February) shows that lightning events could not penetrate high enough to reach the upper troposphere, regardless of whether ERA5 or ICON-CLM simulated meteorology was used. However, during the active lightning period (May, June, July, and August), ERA5 shows approximately twice as many events reaching the upper troposphere than ICON-CLM simulation. Additionally, the number of air parcels associated with each event reaching the upper troposphere was significantly higher in ERA5. However, the slower and more northward transport of air parcels in ERA5 suggests that these parcels follow mesoscale convective systems, which lack the resolution to capture individual deep convective events associated with lightning. This feature is reported in previous studies using ERA-interim as "up-and-over" moisture transport from low-pressure Indo-Gangetic Plain (IGP) to the southwestern Tibetan Plateau (Dong et al., 2017; Barros et al., 2004). Previous studies using Lagrangian transport simulations driven by ERA5 and ERA-Interim data suggest that convective transport in the troposphere is underestimated due to unresolved convective updrafts (Hoffmann et al., 2023; Konopka et al., 2022). To address this limitation, they recommend incorporating additional convection parametrisations into Lagrangian models. In contrast, the km-scale ICON-CLM simulations depict air parcel transport that is more sensitive to individual lightning-producing deep convective events at the grid scale. As a result, most air parcels in ICON-CLM simulations ascend to the upper troposphere within 2–3 hours, with minimal horizontal displacement before reaching the UTLS (4b-d).

### 3.6 Seasonal Analysis

The synoptic wind patterns over the Indian subcontinent vary across seasons. To account for these seasonal variations, we identified events that reached the upper troposphere, categorising 54 events from ERA5 (Figure 4e) and 36 events from ICON-CLM simulations (Figure 4b) based on their respective seasons. The post-monsoon/winter period, was defined as spanning from October 10, 2019, to February 2020, based on the withdrawal of monsoon from the Himalayan mountains and the IGP (India Meteorological Department, 2019). The pre-monsoon season was considered from March 1 to June 14, 2020, based on the onset of the monsoon in the IGP and Himalayan mountains (India Meteorological Department, 2020). Between June 15 and September 2020, the remaining events were grouped into the monsoon season.

Figures 5a and 5b show lightning events during the post-monsoon/winter period, in which air parcels were able to reach near the upper troposphere using meteorology from ERA5 and ICON-CLM simulation, respectively. Those events are located in the western Himalayan region or the southern part of the Himalayan region over IGP. During the winter season, western disturbance originated due to low pressure in the Mediterranean region carries moisture to the upper troposphere; interaction of this system with topography creates clouds and precipitation in the high-altitude region of the Western Himalayan mountains. For winter, ERA5 shows a much higher number of air parcels reaching the upper troposphere (300 hPa) than ICON-CLM simulation for the same events, especially over the Western Himalayan region. However, both ERA5 and ICON-CLM simulation show comparable features of air parcel transport during deep convective (lightning) events over western Nepal and Uttarakhand of India. During winter, Lagrangian tracking from both datasets show that lightning events can moisten the free troposphere up to 300 hPa, but they do not reach higher.

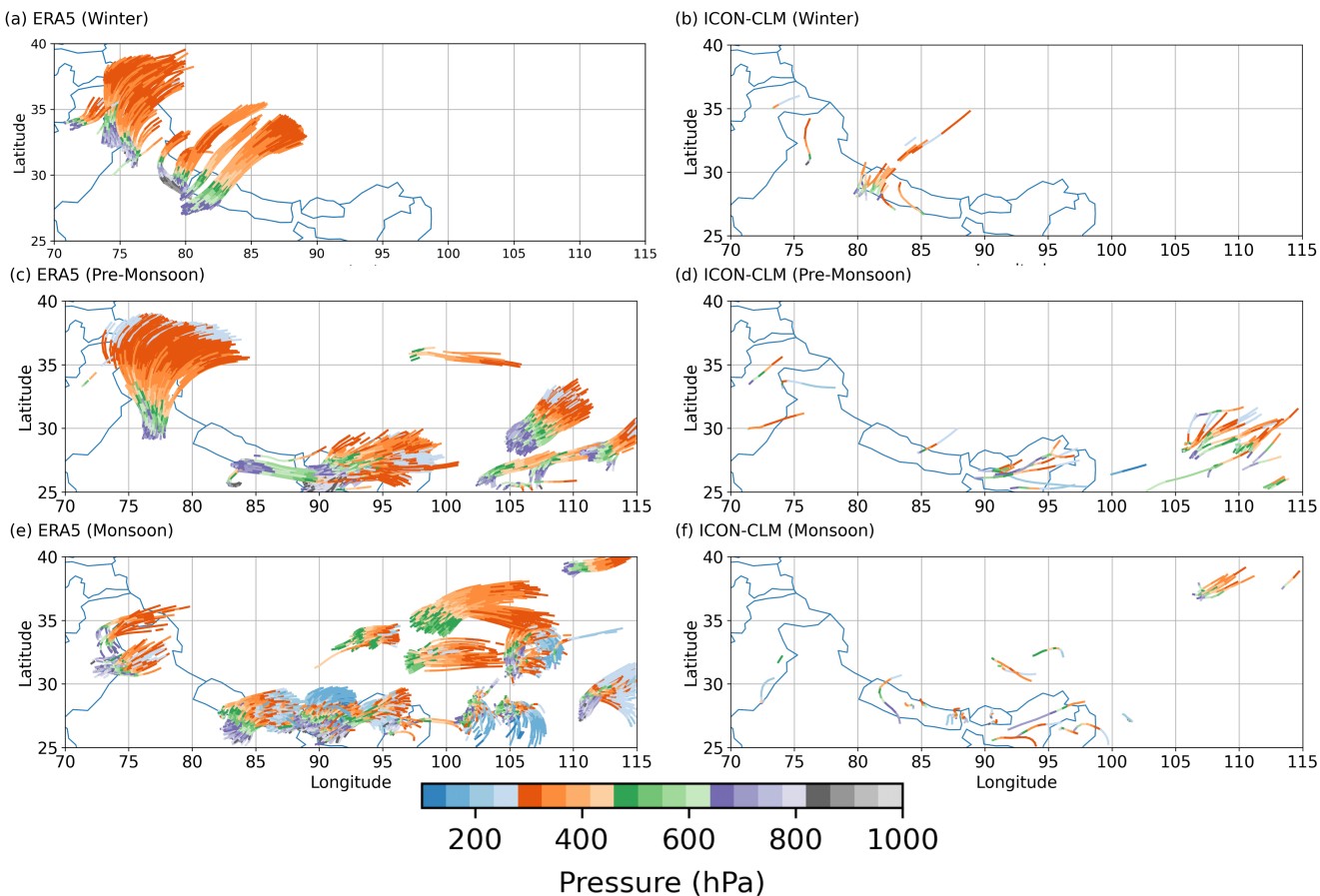

**Figure 5.** The air parcels associated with lightning events reaching the UTLS during different seasons: the first row represents Winter, the second row represents Pre-Monsoon, and the third row represents Monsoon. The first column uses ERA5 meteorology, while the second column uses ICON-CLM simulated meteorology.

During the pre-monsoon period, western disturbances continue to affect northern India, while low-pressure zones form over the Bay of Bengal (Sikka, 2011). These conditions drive deep convection activity over the western and eastern parts of the Himalayan mountain range, respectively. As a result, lightning events are observed, extending from the western Himalayan region through the central Himalayan region to the Brahmaputra Valley (Singh and Ahrens, 2023). Lagrangian tracking of these events using ERA5 meteorology shows slightly higher penetration into the upper troposphere (at a higher altitude)

than in winter, though the transport path remains similar. Air parcels ascend to the free troposphere during deep convection (lightning) events, then cross the Himalayan mountains in the north and are advected horizontally. In contrast, the km-scale ICON-CLM simulation shows stronger vertical updrafts with less horizontal displacement than ERA5. For the events over the Brahmaputra Valley, India and Sichuan Basins, China, both ERA5 and ICON-CLM simulation show updraft to the upper troposphere and then advect eastward. ERA5-driven Lagrangian tracking shows a slower ascent to the upper troposphere

than ICON-CLM simulation for deep convective events. Lagrangian tracking of air parcels during the monsoon indicates that ERA5-driven air parcels frequently ascend to the upper troposphere, sometimes nearing the tropopause. Once air parcels reach the upper troposphere, they remain in the upper troposphere and undergo horizontal advection, eventually moving out of the study domain. Meanwhile, the ICON-CLM-driven air parcels also showed similar features but were quite low in frequency. This indicates that the parametrized scale ERA5 shows a higher frequency of air parcels that reach the upper troposphere than

the km-scale ICON-CLM simulation. Therefore, during monsoon at 200 hPa, ERA5 shows higher water vapour concentration than satellite observations (~100%) and km-scale ICON-CLM simulations (~20%) (Figure 3).

### 3.7   Event Analysis

A climatological and seasonal analysis of the relationship between lightning activity and UTLS moistening, using ERA5 and km-scale ICON-CLM simulations over a year, suggests that ERA5 overestimates moisture in the upper troposphere compared

to ICON-CLM simulation and satellite observations. However, the bias in ERA5 appears to originate from differences in the transport mechanism rather than inaccuracies in representing lower atmospheric moisture content, facilitating the transport of higher water vapour into the UTLS region. For event analysis, we identified the time during each season when ICON-CLM simulated the highest LPI and CP values corresponding to the observed ISS-LIS pass. All lightning events occurring at that specific ISS-LIS pass time were selected for analysis (Figure A3).

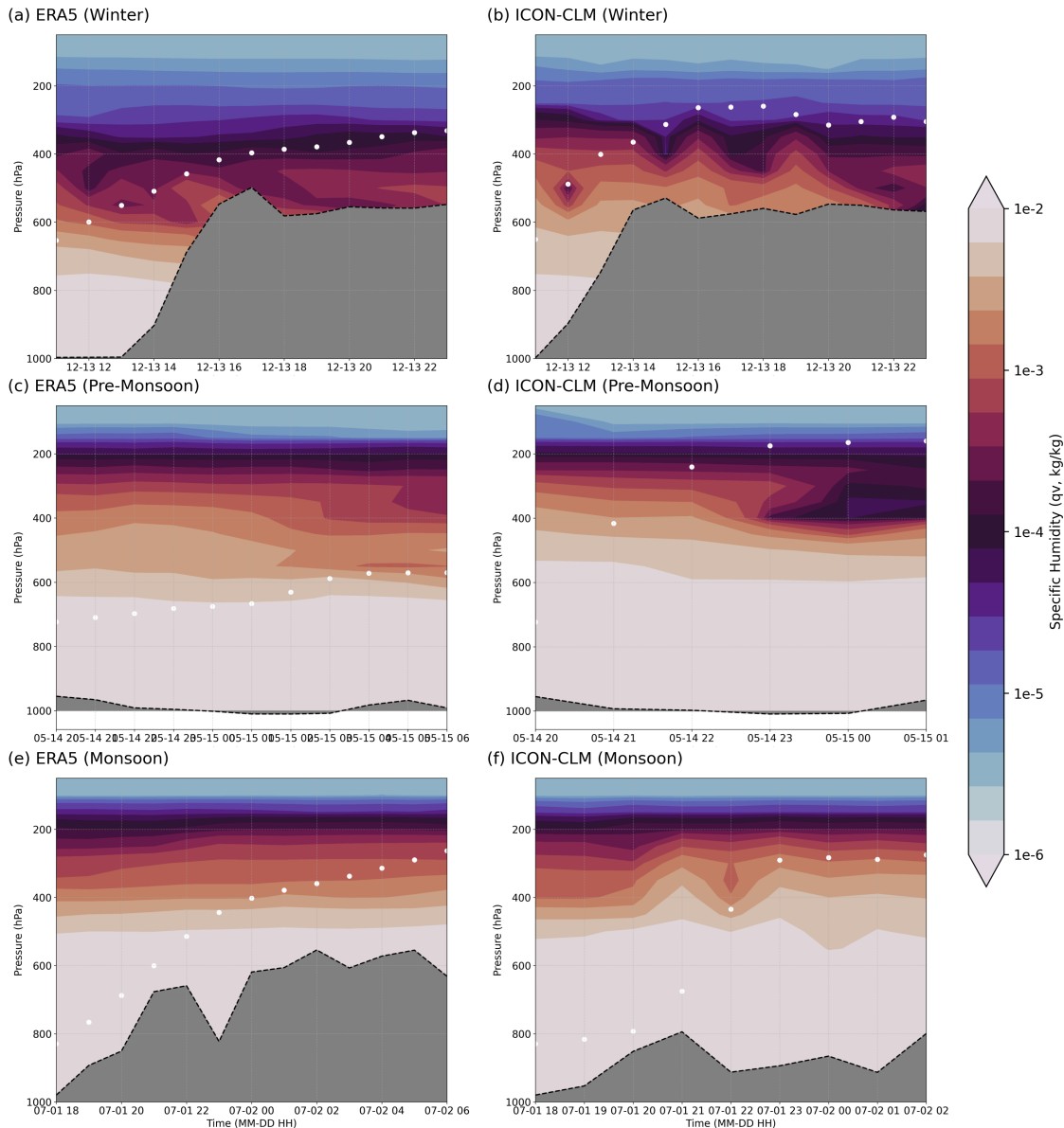

**Figure 6.** The transport path of air parcels (white dot) initiated for Lagrangian tracking for a lightning event reaching the UTLS region during different seasons (Winter (a,b), Pre-Monsoon (c,d), Monsoon (e,f)) for the first 12 hours; the orography follows the path of air parcels. The background shows specific humidity at respective points and time steps. The first column is from ERA5, and the second column is ICON-CLM.

### 3.7.1 Winter Events

We tracked air parcels for all the 14 lightning events during the winter period for the ISS-LIS pass on 13th December 2019 at 12 UTC. ERA5 showed that air parcels from 7 events reached the upper troposphere (above 300 hPa), whereas ICON-CLM indicated only 3 such events.

Figure 6a,b shows the air parcel path (red dot) during a lightning event on 13th December 2019 using ERA5 and ICON-CLM simulated meteorology, respectively. This figure illustrates the position of the air parcel at each forward time step. Since the air parcel's trajectory can vary horizontally and vertically, the orography in both figures varies relative to the air parcel's path. During the winter season, air parcels driven by both ERA5 and ICON-CLM meteorology show northward transport, crossing the Himalayan mountains and reaching the upper troposphere over the Tibetan Plateau. The ERA5-driven air parcels exhibit slower vertical and horizontal transport toward the Tibetan Plateau. In contrast, the km-scale ICON-CLM simulation demonstrates faster and higher transport to the upper troposphere within 3–4 hours. These rapidly transported air parcels appear to be influenced by local disturbances, likely fed by orographic effects or secondary convection produced at km-scale simulation, which is missing with the ERA5.

### 3.7.2 Pre-Monsoon Events

Over 70 lightning events were analysed during the pre-monsoon period for the ISS-LIS observation on 14th May 2020 at 21 UTC. More than 16 parcels (5 near the tropopause) using ICON-CLM meteorology and 22 air parcels (18 near the tropopause) using ERA5 meteorology reached the upper troposphere. Figure 6c,d illustrates one of the lightning events on 14th May 2020 over the eastern Tibetan Plateau. Since ERA5 and ICON-CLM simulation exhibit different air parcel transport patterns during various events, we selected one with high CAPE values for analysis. As in the winter season, ERA5 again shows slow vertical movement toward the north. In contrast, the km-scale ICON-CLM simulation displays rapid transport to the upper troposphere within the first 3 hours of the simulation. ERA5 and ICON-CLM simulation show similar profiles of specific humidity along the air parcel path. Both show elevated specific humidity in the tropopause region compared to the winter season. Figure 5,6 show that during the pre-monsoon period, the upper troposphere is more moisturised than in winter during lightning events.

### 3.7.3 Monsoon Events

Over 70 lightning events were analysed during the monsoon period for the ISS-LIS pass on 1st July 2020 at 19 UTC. A similar number of air parcels from both ERA5 and ICON-CLM simulation were found to reach the upper troposphere during these lightning events, as observed in the pre-monsoon period. However, surprisingly, more air parcels reach near the tropopause with ERA5 compared to ICON-CLM simulation. Additionally, compared to the pre-monsoon period, ERA5 shows more air parcels reaching closer to the tropopause during the monsoon. Figure 6e,f shows the air parcel tracking for a lightning event observed on 1st July 2020, based on ERA5 and ICON-CLM simulations. Both show a moister UTLS compared to the winter season, with moisture levels slightly higher than those observed during the pre-monsoon season. ERA5 shows a similar transport pattern observed during the convective (lightning) events in different seasons, with slow vertical and northward propagation.

Meanwhile, the km-scale ICON-CLM simulation shows fast transport to the upper troposphere over the Tibetan Plateau region along the convective air parcel, which was observed with lightning. Also, ICON-CLM captures features of specific humidity vertical gradients that reflect the influence of orography and convective systems. We can observe in Figure 6e that as the air parcel reaches above 400 hPa, there is a sharp increase in specific humidity; this is also quite evident in winter and pre-monsoon events.

Since all the analysed events were around the Tibetan Plateau region, we additionally examined several events over the Tibetan Plateau. Specifically, we focused on the ISS-LIS pass on 21st August 2020 at 7 UTC, which corresponds to the highest CP and LPI value over the Tibetan Plateau. Out of all the events analysed on 21st August 2020, only one event showed air parcels reaching the upper troposphere (Figure A4). In this event, we can see that the ERA5-driven air parcel shows similar features, as discussed earlier. Meanwhile, the ICON-CLM-driven air parcel shows slow horizontal movement in the free troposphere from the first lightning (convective) event. Then, with consecutive lightning (convective) events, it reaches the upper troposphere close to the tropopause region. Also, vertical changes in specific humidity along air parcels are quite evident. In general, ERA5-driven air parcels show more air parcels reaching the upper troposphere and near the tropopause during the monsoon. Still, specific humidity along the air parcel doesn't show any difference with respect to background concentration. Whereas the km-scale ICON-CLM simulation shows changes in specific humidity along the air parcel path, it also shows a faster ascent compared to ERA5 for most of the events.

The tropopause over the Tibetan Plateau is lower in winter by approximately 50 hPa compared to the monsoon season (Yang et al., 2016). During the winter period, very few air parcels reached near 200 hPa. Despite the lower tropopause, this limited number of air parcels reaching 200 hPa limits the potential for transport into the lower stratosphere for the studied events. These results agree with previous studies which suggest that deep convective events over the Asian Monsoon region can moisten the free troposphere but are limited to reach the UTLS (Uma et al., 2014; Singh et al., 2020). During the pre-monsoon period, the tropopause height is lower than during the monsoon season. However, a similar number of events show air parcel ascent above 200 hPa in both seasons. This suggests that the pre-monsoon period may offer a higher likelihood of penetration into the lower stratosphere during deep convective events over the Third Pole region. This is especially true for late afternoon deep convection, possibly due to diurnal variations in tropopause height (Figure 1), as reported in previous studies (Suneeth et al., 2017), which provides the possibility of mixing of upper tropospheric air with lower stratospheric air.

## 4   Conclusions

Previous studies have reported a correlation between lightning events and upper tropospheric moistening (Price, 2000; Price et al., 2023). The long-term (2003–2013) satellite-based lightning observations, combined with specific humidity data from various sources, reveal a strong correlation in the upper troposphere over the Third Pole region (Figure 2). This analysis suggests that lightning events are a useful indicator of upper tropospheric (~200 hPa) moistening. However, this relationship diminishes in the lower stratosphere (~100 hPa). The Tibetan Plateau and western Himalayan region show a stronger correlation between lightning events and upper tropospheric moisture than other areas within the Third Pole domain.

The daily domain-averaged ICON-CLM simulated LPI strongly correlates with specific humidity in the upper troposphere (Figure 3). However, similar to the climatological observations, this relationship weakens in the lower stratosphere. This suggests that convective events associated with lightning effectively moisten the upper troposphere but are less efficient at transporting moisture to the lower stratosphere. The cold-point tropopause can limit moisture transport into the lower stratosphere through the freeze-drying process (Fueglistaler et al., 2005; Fueglistaler and Haynes, 2005; Smith et al., 2022; Schoeberl and Dessler, 2011). Therefore, the specific humidity in the lower stratosphere tends to show a weaker correlation with lightning activity. Not only LPI and CP, observations from ISS-LIS and WWLLN show a similar trend, in addition they suggest lightning-intense deep convective events to upper tropospheric moistening is more evident during pre-monsoon compared to the monsoon period over the Third Pole region. Lagrangian tracking of air parcels for several lightning-intense deep convective events suggest that up to 10% of such events can directly moisten the upper troposphere. Additionally, the analysis indicates that most air parcels reaching the upper troposphere originate from regions south of the Himalayan Mountains, extending from the western Himalayas to the Brahmaputra Valley along the central Himalayas, and east of the Tibetan Plateau (Figure 4). The ICON-CLM km-scale simulation show a faster air parcel ascent (within 3-4 hours) to the upper troposphere compared to ERA5 ($\geq 8$ hours).

As previous studies suggest, the frequency of lightning events can be a valuable indicator for understanding upper tropospheric moisture, which is quite evident in our analysis using reanalysis and satellite data products (Price, 2000; Price and Asfur, 2006; Price et al., 2023). However, the relationship between water vapour and lightning weakens as we analyse higher altitudes into the upper troposphere and lower stratosphere. ERA5 and km-scale ICON-CLM simulation driven Lagrangian tracking suggest that during lightning events, air parcels can reach up to the upper troposphere but are limited in reaching near the tropopause over the third pole region. Especially, ERA5 shows a higher frequency of air parcels reaching the upper troposphere than the ICON-CLM simulation. Previous studies have reported moist bias in the ERA5 over the upper troposphere region (Ploeger et al., 2013, 2024). A Lagrangian tracking-based study on inter-hemispheric UTLS transport suggest findings similar to this study (Yan et al., 2021). The ERA5 Lagrangian trajectories of air parcels during deep convective (lightning) events appear to follow an 'up-and-over' pattern, as reported in earlier studies (Dong et al., 2017; Barros et al., 2004). These trajectories involve air parcels ascending, crossing the Himalayan mountains, and reaching the upper troposphere over the Tibetan Plateau. Previous Lagrangian tracking studies suggest that ERA5 underestimates deep convective transport, highlighting the need to parametrise vertical updrafts when using ERA5 wind fields in Lagrangian simulations (Hoffmann et al., 2023; Konopka et al., 2022). The Third Pole region—particularly the Himalayan Mountains from west to east, experiences diverse forcing mechanisms for deep convection, such as western disturbances, orographic lifting, and the monsoon. Therefore, it is crucial to carefully tune the convection-related updraft parametrisation in the Lagrangian model to accurately represent extreme convective transport using ERA5 data. In contrast, air parcels simulated using the km-scale ICON-CLM model, which explicitly represents individual convective updrafts and better resolves orographic effects than ERA5, demonstrate a more realistic representation of their paths.

In conclusion, deep convective (lightning) events play an important role in moistening the mid- and upper troposphere, although only a small fraction of air parcels from these events may reach the lower stratosphere over the Third Pole region.

These convective events can transport water vapour and other pollutants to the upper troposphere or near the tropopause, where their fate is governed by synoptic-scale circulation. ERA5 tends to overestimate vertical transport throughout the year, particularly during the monsoon season, resulting in a moisture bias in the UTLS region. In contrast, the km-scale ICON-CLM simulation provides a more realistic representation of transport to the upper troposphere, effectively reducing this moisture bias in the UTLS region. Lagrangian studies using ERA5 need to account for the effects of unresolved convective updrafts

(vertical winds) on transport simulations. These can be better represented using km-scale simulations rather than relying on CAPE-threshold based parametrisations, which may lead to misrepresentation of actual convective transport.

*Code and data availability.* The analysis and post-processing codes are available at https://doi.org/10.5281/zenodo.15090109. Many variables from the year-long ICON-CLM simulation can be accessed through CPTP-CORDEX at http://rcg.gvc.gu.se/cordex_fps_cptp/ (Contribution No. 17). All observational and reanalysis datasets used in this study are publicly available and can be accessed from the sources listed

in the Data and Methodology section. The relevant links and repositories are provided alongside each dataset. Additional simulated datasets, which are not available through CPTP-CORDEX, can be obtained by contacting B.A. (Bodo.Ahrens@iau.uni-frankfurt.de).

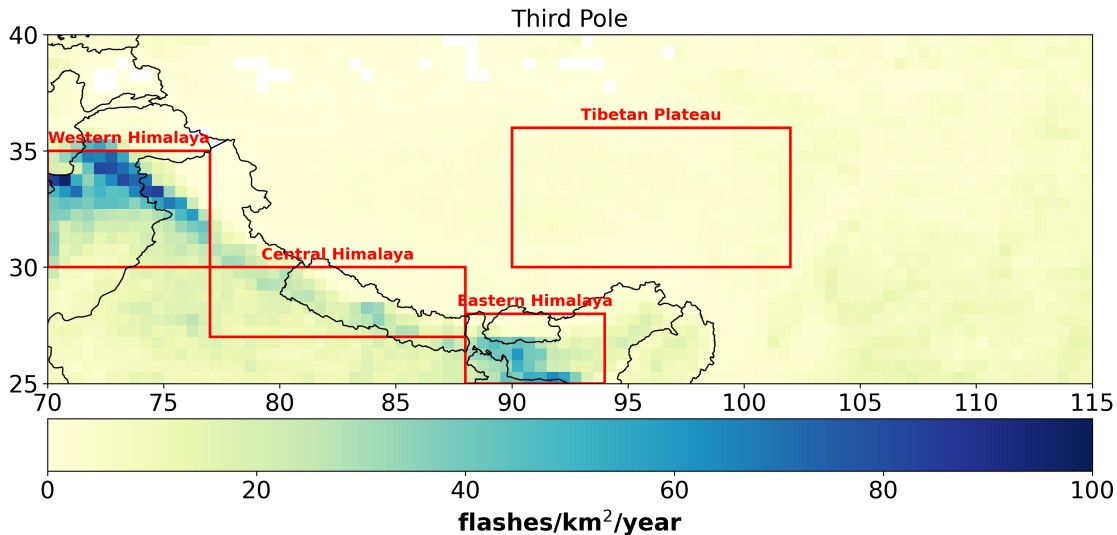

**Figure A1.** TRMM lightning climatology over the Third Pole domain considered in this study. Red boxes indicate regions referred to as the Tibetan Plateau, Western Himalaya, Central Himalaya, and Eastern Himalaya, which are used for lightning climatology analysis.

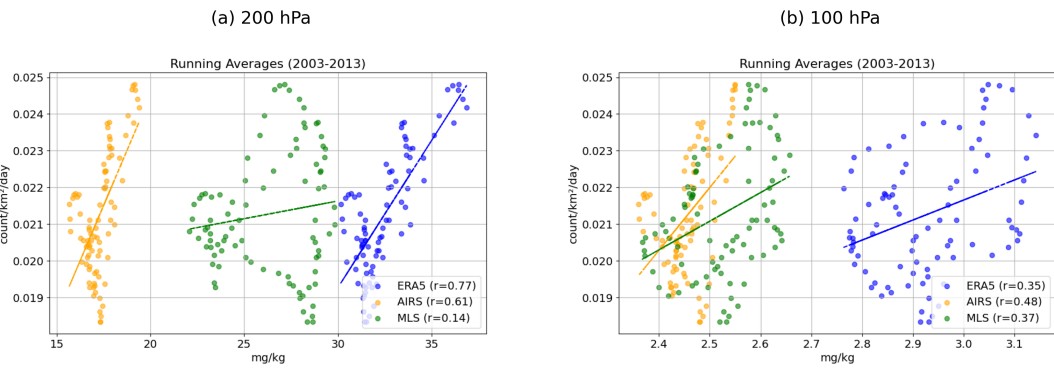

**Figure A2.** Same figure as figure 2 with 12 month running average data.

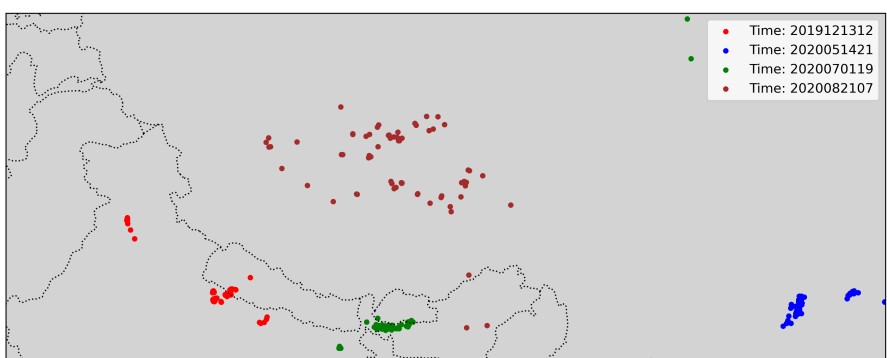

**Figure A3.** Location of the lightning analysed for event study over the Third Pole region, time of events are in YYYYMMDDHH formate

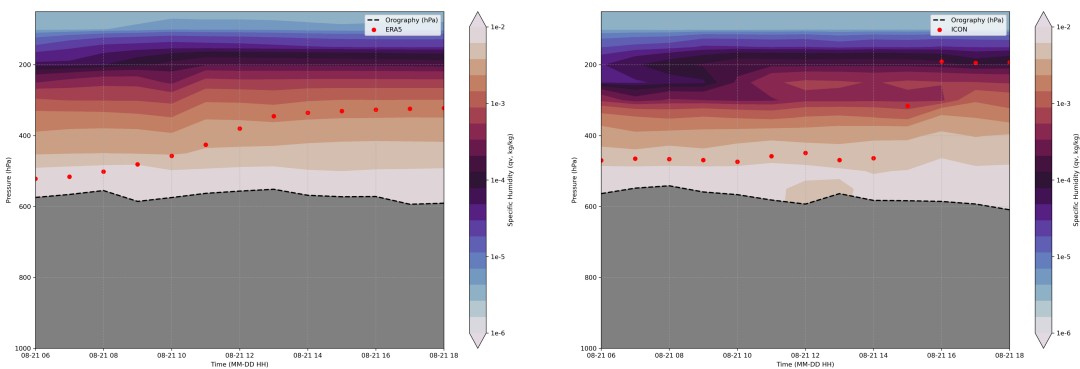

**Figure A4.** The air parcels associated with a lightning event reaching the UTLS region over the Tibetan Plateau. The first column is from ERA5, and the second column is ICON-CLM

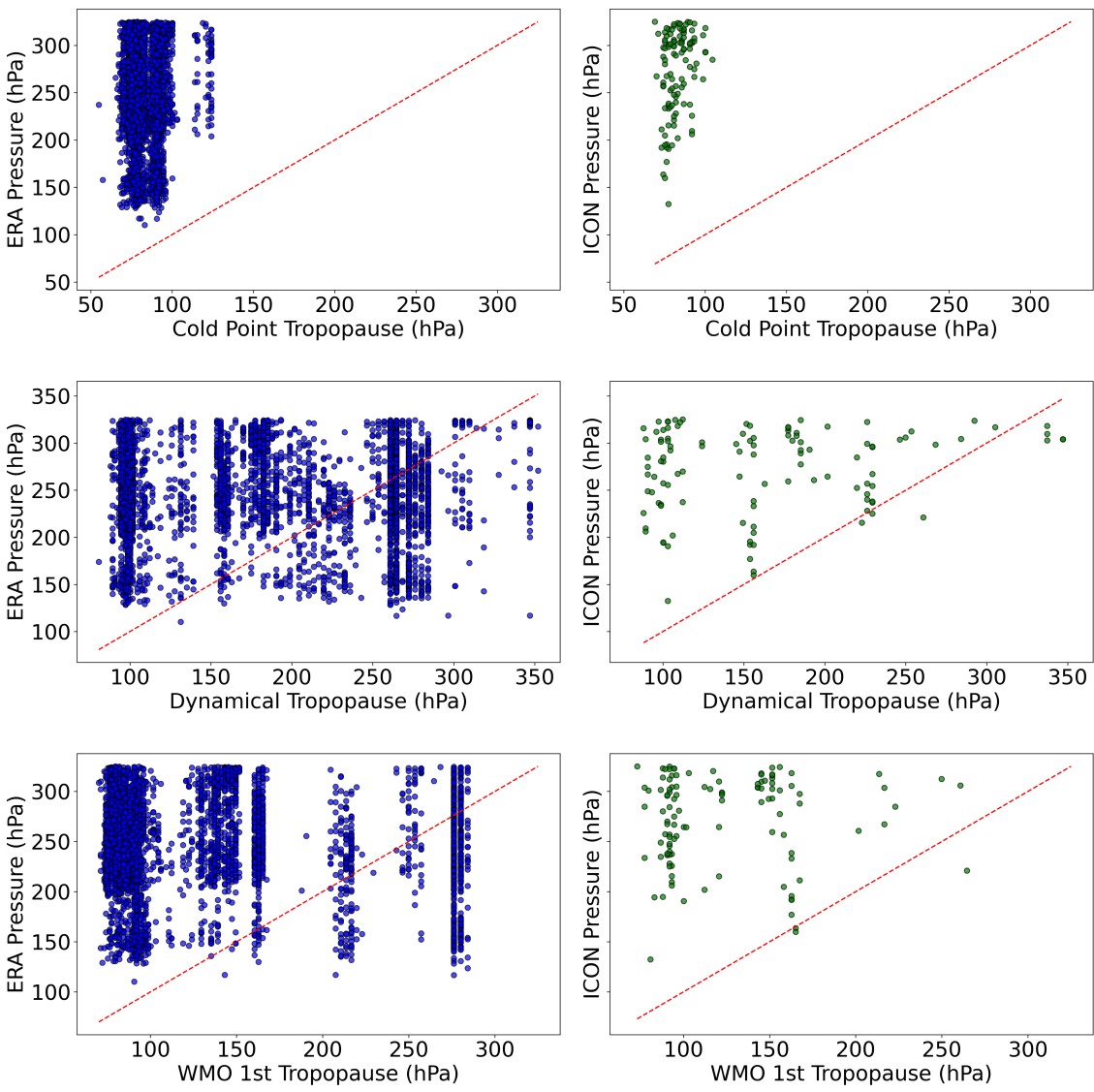

**Figure A5.** Air parcels from ERA5 and ICON-CLM reaching the UTLS region, relative to tropopause height defined by ERA5 (cold point, dynamical, and WMO lapse rate).

**Table A1.** Correlation coefficients for different datasets (ERA, AIRS, MLS) at 200 hPa and 100 hPa pressure levels from 2003-2013 over different regions of the domain.

| Correlation Coefficient | 200 hPa | | | 100 hPa | | |
|---|---|---|---|---|---|---|
| | ERA5 | AIRS | MLS | ERA5 | AIRS | MLS |
| Domain | 0.76 | 0.71 | 0.74 | 0.42 | 0.56 | 0.33 |
| TP | 0.86 | 0.78 | 0.80 | 0.65 | 0.67 | 0.47 |
| WH | 0.61 | 0.61 | 0.66 | 0.47 | 0.63 | 0.40 |
| CH | 0.32 | 0.29 | 0.27 | 0.19 | 0.23 | 0.12 |
| EH | 0.01 | -0.02 | -0.01 | -0.08 | -0.05 | -0.10 |

*Author contributions.* Conceptualization, P.S. and B.A.; methodology, P.S and B.A.; software, P.S.; validation, P.S.; formal analysis, P.S. and B.A.; investigation, P.S.; resources, B.A.; data curation, P.S.; writing—original draft preparation, P.S.; writing—review and editing, P.S. and B.A.; visualization, P.S.; supervision, B.A.; project administration, B.A.; funding acquisition, B.A. All authors have read and agreed to the published version of the manuscript.

*Competing interests.* No competing interests.

*Disclaimer.* No disclaimer.

*Acknowledgements.* This work was funded by the Deutsche Forschungsgemeinschaft (DFG, German Research Foundation) – TRR 301 – Project-ID 428312742. The authors thank Goethe-HLR and DKRZ for providing computational resources.

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
