# Peer review of "Lightning-intense deep convective transport of water vapour into the UTLS over the Third Pole region"

_EGUsphere, 2025_

## Author Comment (AC1)

The study focuses on water vapour transport mechanisms driven by lightning-induced deep convection in the Third Pole region, a global lightning hotspot where UTLS (upper troposphere and lower stratosphere) water vapour exchange significantly impacts climate. The research addresses critical regional and global climatic implications.The combination of multi-source data (TRMM-LIS, ERA5, AIRS, MLS) with km-scale ICON-CLM modeling and Lagrangian tracking creates a robust "observation-reanalysis-model" validation framework, demonstrating methodological advancement.The logical and the results are reasonable. The structure is clear, and the writing is well. However, there are some concerns about this study.

**Response:** We sincerely thank the reviewer for their thoughtful and encouraging feedback. We greatly appreciate your recognition of the scientific significance of our study, particularly in addressing water vapour transport mechanisms in the Third Pole region and their climatic implications through the use of multi-source data and high-resolution ICON-CLM simulations.

1.Lightning datasets (TRMM-LIS, ISS-LIS) span long periods, but the impact of temporal discrepancies (TRMM: 1995–2015; ISS-LIS: 2019–2020) on result consistency requires clarification.

**Response:** We thank the reviewer for raising this important point. We have clarified the use of TRMM-LIS (1998–2015) and ISS-LIS (2019–2020) datasets in the revised manuscript (Data and Methodology section; line numbers 70-80 in revised manuscript). While these datasets span different periods, they share consistent detection capabilities through the similar LIS instrument. TRMM-LIS is used to assess long-term regional lightning patterns, while ISS-LIS supports recent event selection and model evaluation. Since they serve distinct purposes: climatology vs. case-specific validation. The difference in temporal coverage does not compromise the consistency or validity of our results.

2.Incorporate recent (post-2023) observational studies on deep convection in the Third Pole (e.g., ground-based radar or satellite retrievals) to enhance timeliness.
**Response:** We thank the reviewer for this insightful suggestion. We agree that integrating recent (post-2023) observational studies would enhance the relevance of our work. However, we found that post-2023 ground-based radar studies or satellite retrievals specifically focused on deep convection over the Third Pole are currently limited, and most ground-based radar datasets over the Third Pole region remain inaccessible in the public domain.

Although we are involved in a few ongoing collaborative efforts to analyze event-based satellite observations, these studies are still in preparation and are not yet available for citation. Furthermore, the combined analysis of lightning (as a proxy for deep convection) and UTLS water vapor transport representation in high-resolution regional climate models remains largely unexplored over the Third Pole in recent literature. Given these limitations, and the specific focus of our study on evaluating km-scale ICON-CLM simulations for the 2019–2020 period.

3. Correlation analysis (e.g., Table 2) shows the highest lightning-UT water vapour correlation over the Tibetan Plateau (r = 0.78). Whether the region's low lightning frequency (<4 flashes/km²/year) affects statistical significance?

**Response:** We thank the reviewer for raising this important point. While the average lightning density over the Tibetan Plateau is relatively low (<4 flashes/km²/year), our analysis focuses on a spatially extensive region (90–105°E, 30–35°N), which represents the core lightning-active zone within the Plateau (as shown in Supporting Figure 1). Over this large area, flash density accumulates to over 3 million lightning flashes during the TRMM-LIS observation period (1998–2014), ensuring a robust sample size for statistical analysis. Therefore, the observed correlation (r = 0.78) remains statistically significant despite the seemingly low flash density. We have clarified this point in the revised manuscript (Line numbers 154–157 in revised manuscript).

---

## Author Comment (AC2)

Review of "Lightning-intense deep convective transport of water vapour into the UTLS over the Third Pole region", by Prashant Singh and Bodo Ahrens, submitted to Atmospheric Chemistry and Physics (ACP)

This paper investigates the role of lightning-associated convection in transporting water vapour into the upper troposphere and lower stratosphere (UTLS) over the Himalayas and Tibetan Plateau ("Third Pole") region. The authors use lightning data from the Tropical Rainfall Measuring Mission (TRMM-LIS), along with forward trajectories derived from ERA5 reanalysis and high-resolution ICON-CLM simulations, to track moist air masses. The goal of the authors is to assess their contribution to the well-documented water vapour enhancement observed by MLS and ACE-FTS (which are more appropriate than AIRS) within the Asian Summer Monsoon (ASM) anticyclone.

**Response:** We sincerely thank the reviewer for their thoughtful summary and interest in our work. We would like to clarify a key point regarding the study's focus. While we acknowledge the importance of the Asian Summer Monsoon Anticyclone (ASMA) in UTLS water vapour enhancement, our objective was not to directly assess the contribution of lightning-associated convection to the ASMA. Rather, the primary goal of this study was to investigate the representation and mechanisms of deep convection-driven water vapour exchange in the UTLS over the complex and elevated terrain of the Third Pole region.

Specifically, we aimed to evaluate the performance of km-scale ICON-CLM simulations in resolving vertical transport during lightning-associated convection, in comparison with coarser-resolution ERA5 Lagrangian trajectory analysis. The emphasis is on assessing model capabilities in capturing localized and orographically influenced convective processes, rather than attributing large-scale water vapour anomalies within the ASMA.

Since AIRS is a gridded dataset that has been extensively evaluated against radiosonde observations over the Third Pole region, we selected AIRS and MLS for daily and monthly water vapor comparisons. We also considered ACE-FTS(https://doi.org/10.20383/103.01245); however, due to its limited number of occultation observations over our region of interest—both on daily and monthly timescales—it is less suitable for meaningful comparison alongside the other gridded datasets (ERA5, ICON-CLM, AIRS, MLS).

It is well known that, in the tropical lower stratosphere, and also within monsoon systems, the stratospheric water vapour entry values are primarily controlled by the freeze-drying of moist tropospheric air at the cold point tropopause (CPT) (Brewer, 1949; Randel and Park, 2019; Smith et al., 2021; see also the introductions of the Ploeger et al. or Clemens et al. papers cited in your paper). Deep convection that directly crosses the tropopause is also under debate but remains much more difficult to quantify. As a result, all statements related to the stratosphere in this paper remain very qualitative; even the position of the WMO tropopause or the cold point tropopause is completely ignored. In my view, no robust conclusions can be drawn from this study regarding any impact on stratospheric water vapour.

**Response:** We appreciate the reviewer's important point regarding the role of the cold point tropopause (CPT) in regulating stratospheric water vapor and acknowledge the complexity of quantifying deep convection crossing the tropopause. Due to the high computational cost of km-scale simulations and limited storage capacity. We archived temperature and water vapor

profiles at several discrete pressure levels, specifically at 21 levels ranging from 1000 hPa to 50 hPa. Given this coarser vertical resolution, accurately identifying the tropopause height—particularly the CPT—is challenging and may lead to misleading results.

To address this limitation, we adopted a fixed tropopause height of ~100 hPa to represent the lower stratosphere, and ~200 hPa for the upper troposphere, in our air parcel tracking framework. To justify this approach, we analyzed the seasonal characteristics of the tropopause over our study region using ERA5 reanalysis (october 2019 to September 2020). Specifically, we included diurnal and monthly variations of the cold point tropopause height (now presented in Figure 1) and additionally evaluated the WMO's lapse-rate-based tropopause and the dynamical tropopause. Our analysis confirms that the 100 hPa level is a reasonable approximation of the tropopause height over the region and time periods considered.

These additions are now detailed in the revised manuscript (Lines 104–108, 125–141 and Figure 1).

Unfortunately, even regarding the upper troposphere, the findings are quite weak, especially when compared to earlier studies such as Price et al. or Singh and Ahrens (2023). The correlation between enhanced upper tropospheric water vapour and lightning counts is not new, and actually appears more clearly in daily data than in strongly averaged climatologies such as Fig. 1. Even the domain-averaged daily time series (Fig. 2) show large inconsistencies, with unexplained spikes between 15 March and 15 June. The correlation coefficients are actually weakest during the monsoon time (Table 3), the time period when intense thunderstorm activity is expected.

**Response:** We fully acknowledge that the relationship between lightning activity and upper tropospheric moistening has been addressed in earlier studies, such as those by Price et al. (2000, 2006, 2023), where lightning is considered a proxy for deep convective activity and associated vertical moisture transport. However, it is important to highlight key differences in our study.

First, previous works have predominantly relied on global or continental-scale analyses using relatively coarse-resolution data sets (e.g., Schumann resonance, reanalysis, gridded satellite). In contrast, our study focuses on the complex terrain of the Third Pole region and utilizes high-resolution km-scale simulations alongside ERA5 reanalysis. To our knowledge, such a high-resolution investigation of lightning-linked vertical water vapour transport in this region has not been conducted before. We have added lines 179–202 in the discussion to clarify the relevance of Figure 2 (previously Figure 1) and Table 2, and to better distinguish and support our findings from those of previous studies

Second, we agree with the reviewer that the period between 15 March and 15 June shows pronounced lightning peaks in the domain-averaged daily time series (Figure 3, previously Figure 2). This is consistent with findings from Singh and Ahrens (2023) and other studies, which report the pre-monsoon season (April-May-June) as the peak period for lightning activity over the Himalayan region, followed by the monsoon.

Third, while monsoon months are characterized by widespread thunderstorm activity, the weaker correlation between lightning and upper tropospheric water vapour during this time (as

seen in Table 3) is likely due to the dominance of large-scale dynamical systems such as the Asian Summer Monsoon Anticyclone (ASMA), which influences UTLS moistening independently of localized convective events. Thus, the strong moistening observed during the monsoon may not directly align with lightning variability, leading to weaker correlations.

We have revised and expanded the discussion in Lines 212–216 (now referring to Figure 3) to better clarify these points and provide a better interpretation of the seasonal differences in lightning–moisture relationships.

The only truly new contribution, in my view, is the comparison of trajectory behavior between ERA5 and ICON-CLM, as you also highlight in your abstract. While the coarser-meshed (~30 km) convection-parameterized ERA5 data show slow ascent, with air parcels crossing the Himalayas and reaching the upper troposphere over the Tibetan Plateau, the convection-permitting km-scale ICON-CLM model reveals faster vertical and more direct transport for the same events (Figs. 3, 4, and 5).

**Response:** We thank the reviewer for recognizing the novelty of the trajectory comparison between ERA5 and ICON-CLM. Indeed, our primary objective was to illustrate the representation of vertical water vapour transport during deep convective events over the complex terrain of the Third Pole using both km-scale ICON-CLM and coarser ERA5 data. The focus was not on evaluating ASMA dynamics, but rather on highlighting how convection-permitting simulations better capture rapid and direct vertical transport in such regions. We have clarified this intent in the revised manuscript.

However, there is significant potential to improve the presentation and interpretation of these results. For example, the color bar in Fig. 5 is not readable, and the visual contrast makes interpretation difficult. Moreover, interpreting the highest trajectory points in Figs. 3d and 3g as being in the stratosphere seems, at best, an overinterpretation. Without proper reference to the cold point or WMO tropopause, such a claim cannot be supported with confidence.

**Response:** We have revised and improved the figure previously labeled as Figure 5, which now appears as Figure 6 in the revised manuscript. The color bar has been enhanced for clarity, and the overall visual contrast has been adjusted to aid interpretation. In addition, we have imporved all the plots in our manuscript (Figure 1-6).

Regarding the interpretation of the highest trajectory points in Figures 3d and 3g, we acknowledge the concern. In the revised manuscript, we now provide a detailed discussion of the tropopause height (Lines 125–141), supported by Figure 1, which includes reference to the WMO-defined tropopause. Based on this analysis, we interpret that the trajectories likely reach the upper troposphere, approaching the tropopause, and in some cases, potentially near the lower stratosphere. However, we have taken care to avoid any overstatement and now present this interpretation with appropriate context and references to the tropopause structure.

In your conclusions, you attempt to link your findings to the recently identified significant wet bias in the lowermost stratosphere in climate models (Charlesworth et al., 2023; Ploeger et al., 2024). However, your results are strongly confined to the upper troposphere. Furthermore, the

wet bias in ERA5 upper tropospheric water vapour, diagnosed in your paper by comparison with MLS and AIRS data, is also present in the high-resolution ICON-CLM model, which you otherwise describe as more physically realistic in terms of vertical transport along trajectories. This is really confusing.

**Response:** Our primary objective was to assess how frequently air parcels reach the upper troposphere and lower stratosphere during deep convective events over the Third Pole region—one of the highest elevated regions on Earth. While our results are indeed confined mostly to the upper troposphere, we attempted to highlight how the moist bias in this layer may contribute to or reflect the broader wet bias identified in the lowermost stratosphere in climate models (Charlesworth et al., 2023; Ploeger et al., 2024).

In our analysis, ICON-CLM simulates fewer air parcels reaching the upper troposphere compared to ERA5. However, ICON-CLM also shows more rapid and vertically focused transport with reduced horizontal displacement, suggesting that it better captures vertical motions directly linked to deep convection. In contrast, ERA5 seems to reflect a combination of deep convection and other transport processes, potentially overestimating parcel entrainment in the upper troposphere.

The diagnosed wet bias in ERA5 upper tropospheric water vapour—identified through comparisons with MLS and AIRS—also appears in ICON-CLM. We attribute this part to the use of ERA5-based boundary conditions in our ICON-CLM simulations, which may propagate some of the moist bias into the regional model. Nevertheless, the localized and more physically consistent transport in ICON-CLM implies that deep convection is better resolved.

To reduce such biases and improve realism further, especially in the lower stratosphere, future work will focus on global ICON simulations with nested high-resolution domains over the Third Pole. This approach may better resolve cross-tropopause transport and reduce upper-level moist biases.

Given these concerns, I can only recommend rejection of the current version. The manuscript would need to be fundamentally rewritten. Possibly, Figs. 3, 4, and 5 could serve as a starting point for a completely new and more focused version.

**Response:** We thank the reviewer for their feedback and acknowledge the concerns raised regarding the structure and clarity of the manuscript. We would like to emphasize that Figures 3, 4, and 5 (now Figures 4, 5, and 6) indeed represent the core results of our study, highlighting the relationship between lightning activity and upper tropospheric moistening in the Third Pole region.

However, we respectfully clarify that Figures 1 and 2 (now Figures 2 and 3) are not intended as standalone key findings but rather serve to provide important background and a climatological context for the region. To the best of our knowledge, such a regional-scale climatological analysis of lightning and upper tropospheric water vapour over the Third Pole has not been reported before. Including this context is essential for understanding the seasonal variability and interpreting the main findings in a physically consistent framework.

We have significantly revised the manuscript to improve clarity, organization, and the presentation of our key results. We have also added a new figure (Figure 2) to strengthen the study and have refined our discussion throughout the manuscript to enhance focus and readability.

We hope these substantial revisions address the reviewer's concerns and demonstrate the novelty and value of our study within the context of high-resolution, region-specific analysis of deep convection and UTLS moistening.

A few other important points:

**Introduction**

If you want to make claims about the stratosphere, large parts of the introduction would need to be rewritten to reflect the relevant processes and literature more accurately.

**Response:** Thank you for this important comment. Our intention is *not* to make broad claims about stratospheric processes or the stratospheric moisture structure. The manuscript's primary goal is to (i) assess how well ERA5 and a km-scale ICON-CLM simulation represent vertical updrafts during deep convective events over the Third Pole region, and (ii) quantify how often these events effectively reach the lower or upper stratosphere. The focus is thus on the skill of the reanalysis versus the km-scale model in representing convective transport, rather than on a process-level characterization of the stratosphere itself. We have revised the discussion section accordingly to clarify this focus and address the reviewer's concern (Lines 439–445 and 452-454, Figure 1, etc.).

**Singh                                                    2015**

This reference appears to be grey literature and, in my view, should not be used in a peer-reviewed journal submission.

**Response:** Thank for the comment, we have removed this citation.

**Lagrangian                                        Tracking**

There is no proper citation of the Lagrangian trajectory tool used in the study. I strongly recommend using a well-established and widely cited tool such as FLEXPART or MPTRAC for this type of analysis.

**Response:** We thank the reviewer for this valuable suggestion. In this study, we employed a Lagrangian trajectory model developed based on the LAGRANTO framework, which has been successfully applied in several recent studies, including Curtius et al. (2024, *Nature*). While we acknowledge that widely used tools such as FLEXPART and MPTRAC offer more advanced features, our objective was to trace air parcel motion using large-scale 3D wind fields from reanalysis (ERA5) and km-scale ICON-CLM simulation, without adding additional parametrizations related to convection. Our simplified tracking choice allows us to isolate and analyze the effects of convection as represented in the driving datasets themselves (e.g., ERA5 and ICON-CLM), rather than introducing another layer of convective treatment within the

trajectory model that could obscure or complicate interpretation. In doing so, our approach provides a clearer diagnostic of how convection parametrization at the model level affects air parcel pathways in a physically consistent manner.

We have now included the appropriate citations and clarified this rationale in the revised manuscript (Line 118- 124).

---

## Author Response (AR2)

Dear Editor Marc von Hobe,

We have addressed all of the reviewer's comments, and the manuscript has been revised accordingly to incorporate the suggestions while preserving the original intent of the paper. We are confident that the current version now meets the high standards of *Atmospheric Chemistry and Physics* (ACP) in terms of scientific rigor and adequately addresses all the points raised by the reviewer.

For your reference, we have attached a document containing the reviewer's comments (in black) along with our point-by-point responses (in red). We hope that this revised manuscript will be positively considered for publication in *ACP*.

**Reviewer I**

Second Review of "Lightning-intense deep convective transport of water vapour into the UTLS over the Third Pole region", by Prashant Singh and Bodo Ahrens, submitted to Atmospheric Chemistry and Physics (ACP)

The paper has improved significantly. In particular, the focus on the advantages gained through the ICON-CLM simulations—especially in comparison to ERA5—is convincing. However, it is still not a proof that Lagrangian trajectories driven by ICON-CLM winds result in moistening of the stratosphere. The freeze-drying effect along the trajectories is not considered, so only qualitative statements are possible.

**Reply**: We thank the reviewer for appreciating the revised version of our manuscript and for providing valuable feedback to further improve its quality. We fully agree that deep convective events are generally limited to moistening the upper troposphere and only rarely penetrate directly into the lower stratosphere. Our km-scale ICON-CLM simulations indeed show a lower frequency of air parcels reaching near the tropopause, while they frequently ascend into the upper troposphere above 300 hPa—consistent with previous studies.

Another issue remains with the application of the Lagrangian method and, in my opinion, an insufficient description of how it is used to reconstruct stratospheric water vapour transport. Specifically, the freeze-drying process and the ascent of trajectories into the stratosphere above the level of zero convective heating are key to our understanding of water vapour entry into the lower stratosphere. I do not suggest that the authors must implement all these calculations; however, without them, no quantitative statements about moistening of the lower stratosphere can be supported.

A minimum requirement, in my opinion, is to at least **describe these processes** in the introduction, because they form the basis of our understanding of water vapour transport into the stratosphere, and the authors rely on Lagrangian methods for their most important conclusions.

**Reply**: We thank the reviewer for this valuable comment. The main objective of our study is to highlight the performance of the km-scale ICON-CLM simulations in resolving deep convective transport to the UTLS compared to ERA5. Following the reviewer's suggestion, we have revised the Introduction to include a description of key processes governing

stratospheric water vapour transport—specifically, the role of the cold-point tropopause and the freeze-drying effect on lower stratospheric moistening (Line 42-74).

Therefore, I would still recommend **major revision**, due to the following points:

**Abstract:**
Please replace: "result in direct moistening" with "may lead to direct moistening" to better reflect the uncertainty in the findings.

**Reply**: We have revised the sentence in the Abstract accordingly (Line 14).

**Introduction:**
I still miss a section describing the Lagrangian methods used to quantify the transport of water vapour into the stratosphere, particularly by identifying the coldest temperatures encountered along troposphere-to-stratosphere trajectories (Lagrangian cold points, LCPs). See, for example: Fueglistaler et al.; Fueglistaler and Haynes, 2005; Liu et al., 2010; Schoeberl and Dessler, 2011; Smith et al., 2021. This is our current understanding of how moistening of the stratosphere happens.

**Reply**: As per suggestion we have added the introduction section explaning CPT and freeze-drying effect on LS moistuning (Line 43-58).

Here, you could also explain the concept of the level of zero radiative heating—only above this level do trajectories typically begin their slow ascent into the stratosphere, and after freeze-drying at the LCP, air parcels acquire their final amount of water vapour for entry into the stratosphere. Based on this framework, overshooting convection may indeed play a role (see, e.g., Avery et al., 2017; Ueyama et al., 2020; Jensen et al., 2020; Ueyama et al., 2023; Homeyer et al., 2023), and perhaps with ICON-CLM it is possible to represent this process in the model.

**Reply**: As per suggestion we have added the introduction section explaning slow transport process above CPT, as per the suggested literarure (Line 59-70).

**A few other minor but important points:**

– **Figure 1:** Certainly an improvement. However, the cold-point tropopause appears to be closer to 90 hPa than 100 hPa. Also, when using Lagrangian modeling, the Lagrangian cold point is more relevant than the Eulerian cold-point tropopause. This further supports the point that your statements regarding transport into the stratosphere remain rather qualitative.

**Reply**: We thank the reviewer for the helpful observation. In our analysis, we considered the cold-point tropopause to be approximately 100 hPa, as maximum height varies from about 88 hPa at 26° N to 110 hPa at 40° N (as Figure 1a and b). The Lagrangian trajectories (specially for ERA5) show vertical transport above the Himalayas and subsequent advection toward the Tibetan Plateau. We agree that, with this tropopause height, our discussion of moistening in the UTLS region remains qualitative primarily representing upper tropospheric rather than lower stratospheric moistening.

– **Figure 4:** This figure is not well explained. Panels (b) and (e) cannot be the initial points—they are clearly not the same. I think these panels show trajectory positions after

some time has passed. Here, by the way, the results of ICON-CLM (localized convection, strong vertical motion, and short timescales), in comparison to ERA5-driven calculations, are presumably more physically realistic. In my opinion, this is the strongest part of your paper.

**Reply**: We thank the reviewer for this valuable observation. Indeed, Panels (b) and (e) for ICON-CLM and ERA5 differ because 100 air parcels were released within each $1° \times 1°$ grid box around the lightning location. The convective updraft strength and activated grid points can vary between the two datasets (Lines 306–307). Consequently, the initial point of air parcels that reach 300 hPa or higher may differ for the same event in ERA5 and ICON-CLM simulation, even within the same 1° grid box.
* * *
**Final remarks:**
I think the lightning-based approach is certainly innovative and shows potential as a new source of information related to the UTLS water vapour budget. Because of this, the work is worth publishing. The points mentioned above can certainly be clarified with some effort.

**Reply**: We sincerely thank the reviewer for the kind appreciation and insightful comments that helped us improve our manuscript. We have made significant revisions to the Introduction, particularly addressing the reviewer's suggestions regarding the cold-point tropopause (CPT) and the freeze-drying effect. In addition, we have added a new supporting figure A4 showing the highest altitude reached by air parcels with respect to various respective tropopause heights, and discussd the reulsts (Line 323-333).

I look forward to your kind response.

Prashant Singh